# Apicobasal RNA asymmetries regulate cell fate in the early mouse embryo

Azelle Hawdon [1], Niall D. Geoghegan [2], Monika Mohenska[3,4,5], Anja Elsenhans[6], Charles Ferguson[7], Jose M. Polo [1,3,4,5], Robert G. Parton [7,8] & Jennifer Zenker [1] ✉

The spatial sorting of RNA transcripts is fundamental for the refinement of gene expression to distinct subcellular regions. Although, in non-mammalian early embryogenesis, differential RNA localisation presages cell fate determination, in mammals it remains unclear. Here, we uncover apical-to-basal RNA asymmetries in outer blastomeres of 16-cell stage mouse preimplantation embryos. Basally directed RNA transport is facilitated in a microtubule- and lysosome-mediated manner. Yet, despite an increased accumulation of RNA transcripts in basal regions, higher translation activity occurs at the more dispersed apical RNA foci, demonstrated by spatial heterogeneities in RNA subtypes, RNA-organelle interactions and translation events. During the transition to the 32-cell stage, the biased inheritance of RNA transcripts, coupled with differential translation capacity, regulates cell fate allocation of trophectoderm and cells destined to form the pluripotent inner cell mass. Our study identifies a paradigm for the spatiotemporal regulation of post-transcriptional gene expression governing mammalian preimplantation embryogenesis and cell fate.

The mammalian preimplantation embryo undergoes dramatic transcriptional and morphological changes to gradually form a complex multi-cellular structure constituted of diverse cell lineages[1]. Initially transcriptionally quiescent, the zygote progressively activates its genome, replacing maternally inherited global RNA transcripts, concomitant with the modulation of transcription factor dynamics[2,3], histone modifications[4] and non-coding RNA expression[5], thereby initiating intercellular heterogeneities. However, it is not until the late 8-cell stage when the first morphological differences in mouse preimplantation embryos emerge, as blastomeres compact and then polarise to establish an apicobasal axis[6,7]. Subsequently, an initial series of oriented cell divisions, followed by apical constriction, directs the physical separation of outer and inner cells at 16-cell stage,

accompanied by the expression of lineage-specific genes, for instance *Cdx2* and *Oct4*, respectively[1,8]. The polarised outer trophectoderm cells, which will give rise to extra-embryonic tissues of the placenta, progressively create a sealant along cell-cell junctions by the expansion of cortical apical actin rings and a polarised intracellular microtubule cytoskeleton, enabling the formation of the blastocyst[9].

A second wave of symmetric-asymmetric cell divisions occurs as the embryo reaches 32-cell stage, completing the first cell lineage decision during mammalian preimplantation embryogenesis[1]. Despite little knowledge on the mechanism of cellular regulation, the inner cells emerging at 16-cell stage contribute predominantly to the pluripotent epiblast, whereas the primitive endoderm is mostly composed of inner cells originating at 32-cell stage, modulated, for instance, by

[1]Australian Regenerative Medicine Institute, Monash University, Clayton, Victoria, Australia. [2]Walter and Eliza Hall Institute of Medical Research, Department of Medical Biology, University of Melbourne, Melbourne, Victoria, Australia. [3]Department of Anatomy and Developmental Biology, Development and Stem Cells Program, Monash Biomedicine Discovery Institute, Clayton, VIC 3800, Australia. [4]Adelaide Centre for Epigenetics, University of Adelaide, Adelaide, South Australia, Australia. [5]South Australian immunoGENomics Cancer Institute, University of Adelaide, Adelaide, South Australia, Australia. [6]Department of Biology, University of Duisburg-Essen, Essen, Germany. [7]Institute for Molecular Bioscience, University of Queensland, Brisbane, Queensland, Australia. [8]Centre for Microscopy and Microanalysis, University of Queensland, Brisbane, Queensland, Australia. ✉e-mail: Jennifer.Zenker@monash.edu

the transcription factor *Sox2*[3]. Approaching the final stage of pre-implantation embryogenesis, pluripotent apolar epiblast cells are sorted spatially from the epithelial-like monolayer of primitive endoderm cells, facing the blastocoel[2].

For the correct adoption of cell fate and function, a tight control of gene expression in space and time is pivotal[10,11]. How a mammalian embryo coordinates signals cooperatively across all blastomeres, while subsets of blastomeres undergo diverse fate transitions to specific lineages remains an open question. Differential RNA localisation correlated with changes in morphology during early embryogenesis, plays a key role in asymmetric cell fate specification, polarisation and embryonic patterning[12], as showcased by seminal discoveries in non-mammalian species. Pioneering work in *Drosophila melanogaster* led to the discovery of morphogen gradients determining anterior-posterior spatial body axis segmentation[13]. Such subcellular heterogeneities are established by the microtubule cytoskeleton, to guide *bicoid* RNA anteriorly which is then translated into an anterior-to-posterior protein gradient at the blastoderm stage[14]. It was also in *D. melanogaster* that advances in high-resolution single-molecule imaging enabled the visualisation of real-time RNA dynamics in intact living 3D embryos[15], using techniques adopted from cell culture studies[16].

In mammals, the application of global transcriptomics and more recently single cell transcriptomics have provided invaluable insights into the dynamic changes of transcripts at consecutive stages of embryo development[17,18]. However, these techniques rely on isolated RNA from dissociated cells and thus do not provide precise temporal or spatial gene expression maps required to dissect the direct coordination of transcriptional and morphological changes on the cellular or subcellular level. Differential subcellular spatial RNA organisation might be a key factor contributing to lineage specification in non-mammalian species, but whether RNA localisation occurs during asymmetric-symmetric cell divisions in mammals remains limited to asymmetrically localised *Cdx2* mRNA transcripts in the late 8-cell stage mouse embryo[7,19].

Here, we use live imaging to dissect the spatiotemporal dynamics of global RNA transcripts during mouse preimplantation development. We reveal an asymmetric RNA enrichment at the basal regions of outer blastomeres at the 16-cell stage and determine the translational activities of blastomeres as they acquire pluripotent or differentiating cell fates. These findings provide potential structural determinants for cell fate, closely associated to gene regulation, and serve as a framework for how subcellular asymmetries in RNA distribution and translation capacity synergistically contribute to cell plasticity during the development of the mammalian embryo.

## Results

### Apicobasal RNA asymmetries established at 16-cell stage

To investigate the spatiotemporal distribution of global RNA transcripts during mammalian preimplantation embryogenesis, we imaged live mouse embryos, co-labelled with the red fluorescent RNA-targeted selective styryl probe F22[20] and Membrane-GFP, from 2-cell to blastocyst stage (Fig. 1a–c and Supplementary Fig. 1a). The RNA specificity of the dye was confirmed by RNase digest of preimplantation mouse embryos (Supplementary Fig. 1b) as seen in other systems[20–23]. From the 2- to 8-cell stage, RNA was widely dispersed as puncta-like foci throughout the entire cytoplasm of blastomeres, with no apparent spatiotemporal patterning (Fig. 1a, f and Supplementary Fig. 1a, c). The distance between individual RNA foci was similar in the entire volume of unpolarised 8-cell stage blastomeres with $3.75 \pm 0.16 \, \mu m$ in areas without cell-cell contacts and $4.05 \pm 0.18 \, \mu m$ in areas surrounded by neighbour cells (Fig. 1d, g).

As the embryo enters 16-cell stage, RNA foci progressively accumulated into larger clusters throughout the cytoplasm of outer blastomeres (Fig. 1b, f and Supplementary Fig. 1d). Strikingly, in late 16-cell stage embryos, RNA foci in outer blastomeres adopted an asymmetric localisation proximal to the basal membrane (Fig. 1c, Supplementary Fig. 1e and Supplementary Movie 1). Although some RNA foci remained within the apical region, analysis of the RNA fluorescence expression intensity across outer blastomeres showed a gradual increase of RNA from apical to basal (Fig. 1f and Supplementary Fig. 1e). Furthermore, the distance between RNA foci was reduced to $2.01 \pm 0.07 \, \mu m$ basally compared to $4.01 \pm 0.18 \, \mu m$ apically (Fig. 1g), supporting the clustered state of basal RNA foci in our live imaging data (Fig. 1c). These results were confirmed using the alternative green fluorescent RNA dye E36[20] (Supplementary Fig. 1f).

Inner blastomeres, however, maintained a dispersed but clustered cytoplasmic RNA localisation pattern with $3.5 \pm 0.13 \, \mu m$ between RNA foci, similar to early 16-cell stage blastomeres (Fig. 1c, g). Accordingly, no significant difference in RNA fluorescence intensity profiles irrespective of the measurement plane was observed (Fig. 1h). Yet, the total RNA fluorescence intensity in outer and inner blastomeres, normalised to cell volume, was similar (Fig. 1i and Supplementary Fig. 1g), indicating that the location but not the overall amount of RNA changes as cells adopt different fates.

At blastocyst stage, when the blastomeres are committed to a trophectoderm or pluripotent inner cell mass fate, RNA clusters re-assume a widely spread distribution throughout the entire cytoplasm (Supplementary Fig. 1a). Thus, our results suggest a subcellular mechanism which drives a transient asymmetric RNA localisation in outer but not inner blastomeres at 16-cell stage, which might be essential for cell fate allocation and mouse preimplantation embryogenesis.

### Polarised microtubule network guides apical-to-basal RNA transport

Directed transport of RNA is facilitated by the guidance of the microtubule cytoskeleton as demonstrated in differentiated cells and non-mammalian embryos[10,19]. Thus, we sought to explore the cytoskeletal mechanisms aiding the asymmetric RNA localisation to the basal membrane in outer blastomeres of 16-cell stage mouse embryos. Throughout the 16-cell stage, preimplantation mouse embryos establish an apical-to-basal expanding polarised microtubule network, inhibiting an opposing cortical actin flow which causes the formation of apical actin rings[9].

In live early 16-cell stage embryos expressing fluorescently-tagged Microtubule-associated protein 2c (eGFP-MAP2c) and Utrophin fused to teal fluorescent protein (mTFP-Utrophin)[24] to label filamentous actin, we observed an increased accumulation of RNA foci inside the denser apical microtubule network[9] (Fig. 2a). Thus, we next followed the re-localisation of RNA in real-time as the embryo enters 16-cell stage ($t = 0$ min) until mid to late 16-cell stage ($t = 6$ h), simultaneously with the expansion of the apical microtubule network (Fig. 2b, Supplementary Fig. 2a and Supplementary Movie 2). RNA accumulated into small clusters which progressively became larger and gathered adjacent to the actin ring near cell-cell junctions in outer blastomeres (Fig. 2b, Supplementary Fig. 2a and Supplementary Movie 1), an area highly enriched in growing microtubule plus ends[9]. As the embryo progresses from early to mid-16-cell stage the microtubule network expands, and at the same time RNA moves basally ($t = 1$ h 22 min to 6 h) (Fig. 2b, Supplementary Fig. 2a and Supplementary Movie 1, 2), suggesting a highly spatiotemporally regulated microtubule-dependent transport of RNA at the time of the first embryonic cell fate decision.

All outer blastomeres of 16-cell stage embryos established an enrichment of RNA near the basal membrane at mid to late 16-cell stage, defined as ~6 to 8 h following the last blastomere to divide from the 8- to 16-cell stage, and within ~1 h of each other (Fig. 2b, c). After its establishment, the apical-to-basal RNA gradient was maintained for the remaining duration of the 16-cell stage, until the next cell division (Supplementary Fig. 2b). Furthermore, total RNA fluorescence

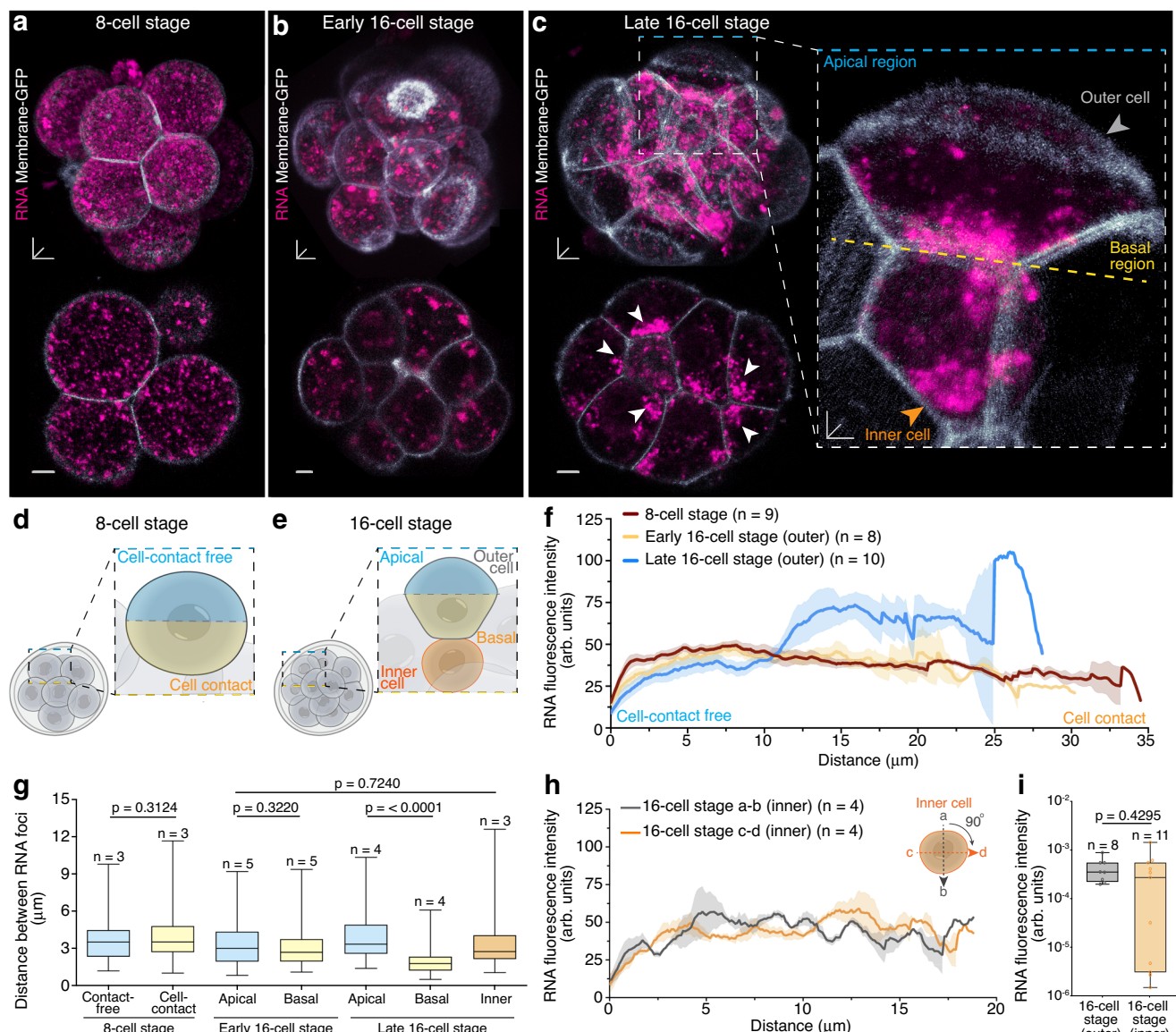

**Fig. 1 | Apical-to-basal RNA gradient in outer cells of live 16-cell stage mouse embryos. a–c** Live mouse embryos labelled with Membrane-GFP show RNA localisation patterns in 3D (top) and 2D (bottom) view at **a** 8-cell stage, **b** early 16-cell stage and **c** late 16-cell stage, which includes close-up view of asymmetric RNA localisation at basolateral regions (white arrowheads in 2D view; yellow dashed line in inset). **d** Schematic representation of an 8-cell stage embryo; blastomeres have a cell-contact free portion (surface facing the external perivitelline space; light blue) and a cell-contact portion surrounded by neighbour blastomeres (yellow). **e** Schematic representation of an 16-cell stage embryo illustrating inner (orange) and outer blastomeres, with apical (light blue) and basal (yellow) portion. **f** RNA

intensity profiles in 8-, early and late 16-cell stage embryos. Data are presented as mean ± SEM; thick lines indicate mean, transparent shadows depict SEM. **g** Distance between RNA foci in 8-, early and late 16-cell stage embryos. Data are presented as mean ± SEM; thick lines indicate mean, transparent shadows depict SEM. **h** RNA intensity profiles in inner cells. Data are presented as mean ± SEM; thick lines indicate mean, transparent shadows depict SEM. **i** RNA fluorescence intensity between outer and inner cells at late 16-cell stage. Box plots display minimum, lower quartile, median, upper quartile and maximum. Mann-Whitney two-tailed t-tests were used to identify statistical differences. Scale bars 3D and 2D embryos 10 μm; inset 5 μm. Source data are provided as a Source data file.

intensity of outer blastomeres was comparable between early and later 16-cell stages (Supplementary Fig. 2c). In summary, our data demonstrate that the microtubule cytoskeleton expands concurrently with a basally directed transport of RNA in early 16-cell stage outer blastomeres.

To further resolve the microtubule dependency of RNA transport, we performed high spatial resolution imaging which revealed a close alignment of smaller RNA foci and basal RNA clusters with the microtubule cytoskeleton at late 16-cell stage (Fig. 2d). Subsequent high temporal imaging at ~1 s intervals demonstrated the directional movement of smaller RNA foci from apical to basal along the microtubule filaments (Fig. 2e and Supplementary Movie 3). To gain further

insight into the directional movement of RNA, we performed real-time tracking of RNA foci in multiple outer blastomeres of living 16-cell stage embryos. The trajectories revealed two subpopulations of distinct RNA dynamics (Fig. 2f and Supplementary Movie 4). A population of more dynamic RNA foci which covered greater displacements of up to 21.30 μm (Fig. 2f and Supplementary Fig. 2d. These RNA foci travelled predominantly in an apical-to-basal directionality at an angle of 81.11° ± 6.93° to the actin ring border (Fig. 2f). Conversely, less mobile RNA foci remained predominantly basally, dwelling with a maximal displacement of <3.5 μm (Supplementary Fig. 2d).

Global loss of microtubules following nocodazole treatment in early 16-cell stage embryos prevented the movement of RNA towards

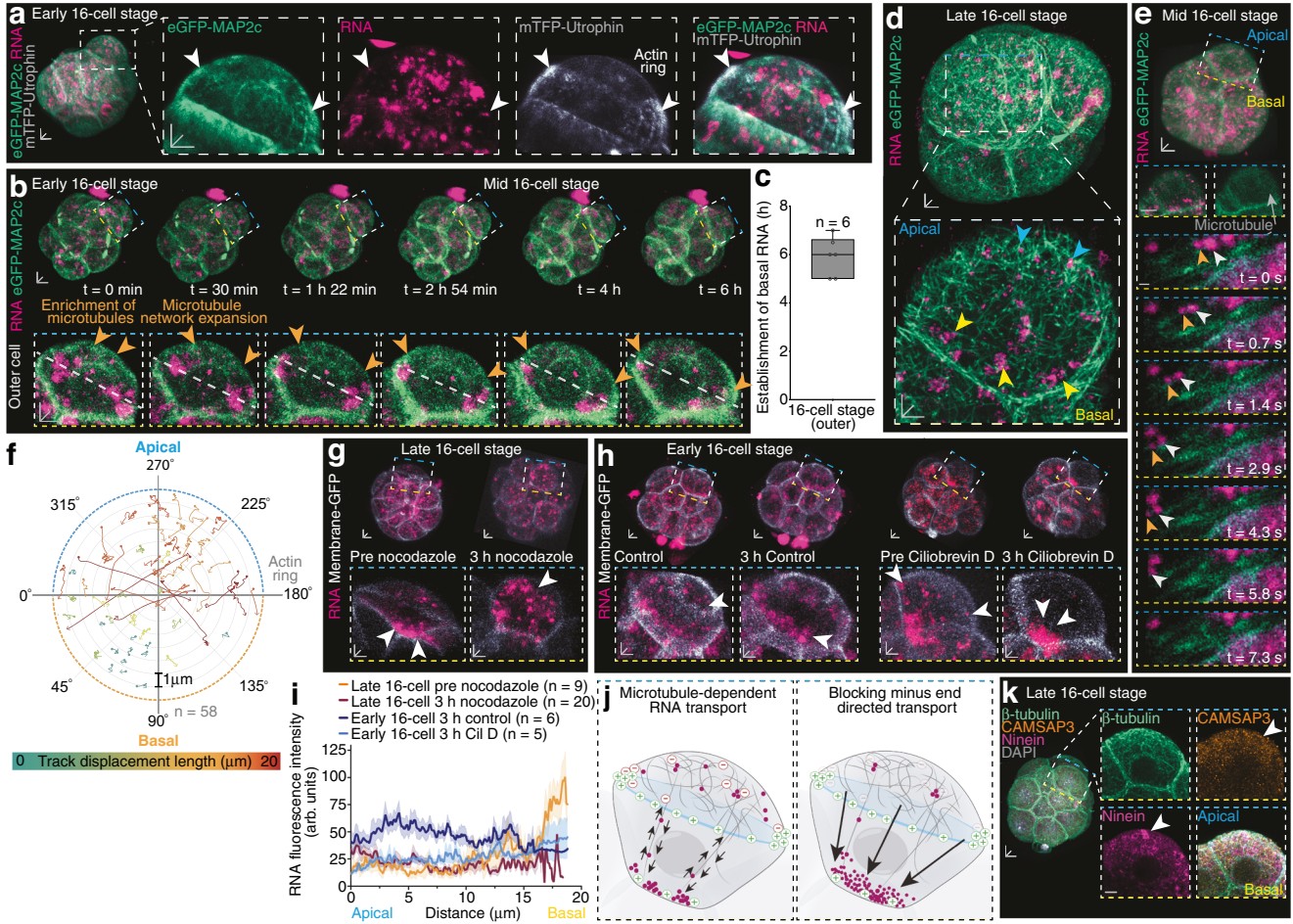

**Fig. 2 | Expanding polarised microtubule cytoskeleton directs accumulation of basal RNA clusters during 16-cell interphase. a** RNA distribution at the denser apical microtubule network (eGFP-MAP2c) underlying the actin ring (mTFP-Utrophin) (white arrowheads). **b** At the onset of 16-cell stage ($t = 0$ min), clustered RNA foci distribute throughout outer blastomeres, but progressively move basally, simultaneously with the expansion of the dense microtubule network (orange arrowheads) and presumptive actin ring border (white dashed line) ($t = 30$ min until 6 h). Entire embryo (top) and individual outer blastomere (bottom) shown. **c** RNA reaches basal region ~6 h into 16-cell stage interphase. n = embryos. Box plots display minimum, lower quartile, median, upper quartile and maximum. Individual data points are overlaid. **d** Apical RNA foci (blue arrowheads) and basal RNA clusters (yellow arrowheads) are aligned along microtubule filaments, **e** which guides the apical-to-basal transport of RNA (white and orange arrowheads). **f** Circular plot shows tracks of directed RNA transport from apical to basal regions in outer

blastomeres of 16-cell stage embryos. Trajectories are statistically colour-coded according to displacement length (μm/s) at angles relative to the longitudinal axis of actin ring (grey line). Circle depicts foci start point, direction of transport depicted by arrowhead. **g** Depolymerisation of microtubules with nocodazole in late 16-cell stage embryos re-distributes RNA apically (white arrowheads). **h** Blocking microtubule minus end-directed transport with Ciliobrevin D intensifies RNA asymmetries (white arrowheads). **i** RNA intensity profiles following nocodazole and Ciliobrevin D (Cil D). Data presented as mean ± SEM; thick lines indicate mean, transparent shadows depict SEM. **j** Schematic representation of polarised microtubules directing RNA transport (left), fortified when microtubule minus end trafficking is blocked (right). **k** Immunofluorescence of late 16-cell stage embryos stained for microtubule minus end markers CAMSAP3 and Ninein, enriched apically (white arrowheads). Scale bars 10 μm; insets 5 μm; except insets in **e** 1 μm. Source data are provided as a Source data file.

the basal membrane and instead RNA remained dispersed throughout outer blastomeres (Supplementary Fig. 2e, f). The same experiment performed at late 16-cell stage, when RNA was enriched asymmetrically at the basal membrane, resulted in the release of RNA, leading to a more dispersed RNA distribution throughout the cytoplasm of outer blastomeres (Fig. 2g, i). By preventing microtubule minus end-directed transport using dynein inhibitor Ciliobrevin D in early 16-cell stage embryos, RNA reached the basal membrane 2.5 h faster than control embryos (Fig. 2h, i). Furthermore, RNA displayed increased clustering following Ciliobrevin D treatment (Fig. 2h).

These results suggest that microtubule plus end transport might be triggering the accumulation of basal RNA clusters. Furthermore, microtubule minus end transport might be directed apically and responsible for the retention of the smaller apical RNA foci (Fig. 2j). Accordingly, we observed a higher expression of microtubule minus end markers CAMSAP3 and Ninein at the apical side of 16-cell stage

outer blastomeres (Fig. 2k). These results support our finding that the localisation of RNA is mediated by microtubule filaments and demonstrate that both the orientation of microtubule minus and plus ends and the basally directed growth of microtubule filaments are critical for the spatiotemporal sorting of RNA transcripts in the preimplantation mouse embryo.

## Lysosomes co-traffic RNA transcripts basally

Targeted transport of membrane-less RNA molecules along microtubule filaments often requires the association with mobile and membrane-bound organelles. In other systems, lysosomes were identified to serve as vehicles for RNA trafficking[25]. To examine whether a similar mechanism occurs in preimplantation embryos, we explored the spatiotemporal dynamics of lysosomes using the Endo/Lysosomal-associated membrane protein LAMP1[25]. Live imaging of preimplantation mouse embryos from the 2- to 8-cell stage revealed

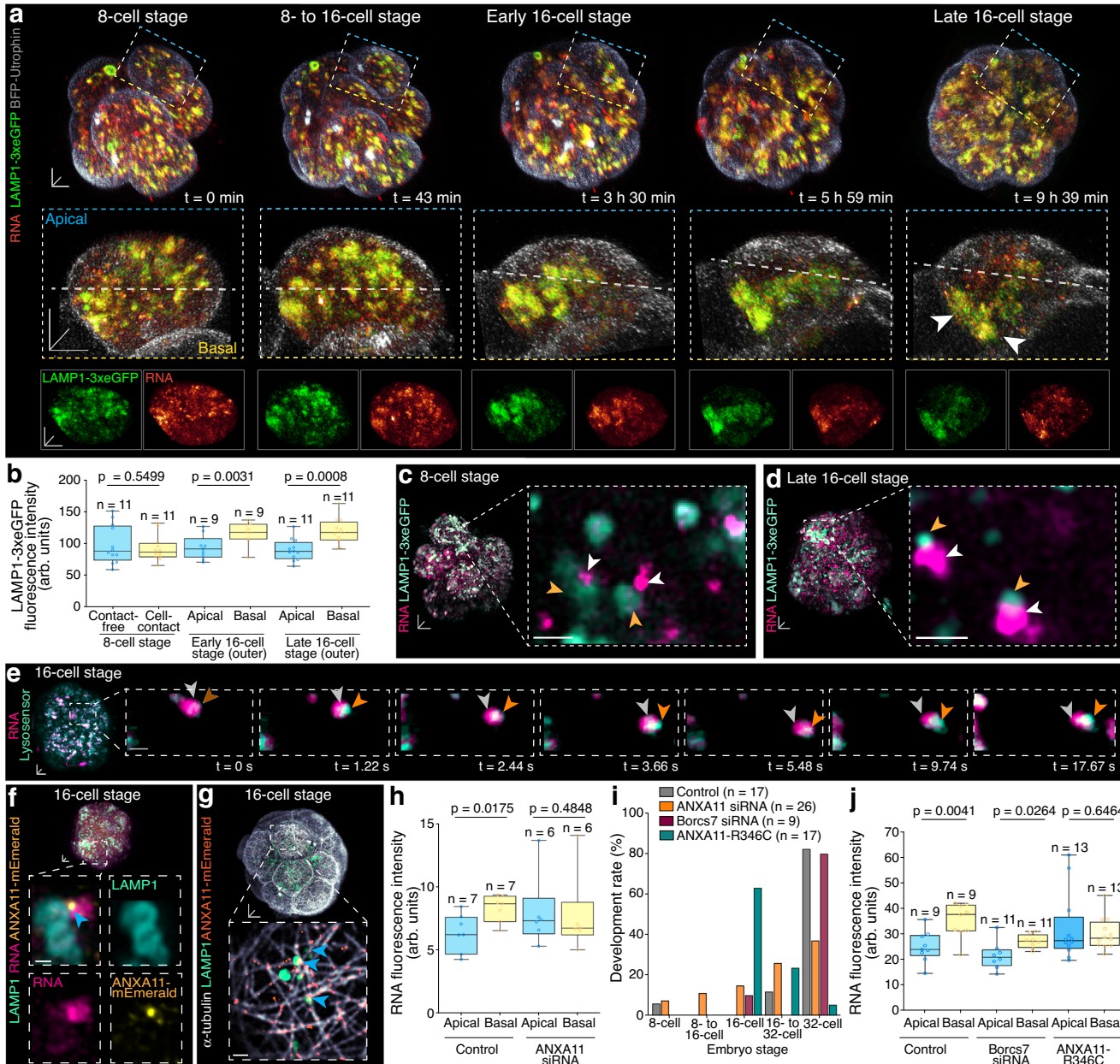

**Fig. 3 | Microtubule-mediated lysosome trafficking shuttles RNA. a** 3D view of dividing 8- to 16-cell stage embryo labelled with RNA, LAMP1-3xeGFP and BFP-Utrophin (top). Merged (middle) and single channels (bottom) show segmented and masked outer blastomere (insets) with apical-to-basal border indicated (dashed line). White arrowheads indicate basal co-enrichment. **b** Quantification of LAMP1-3xeGFP fluorescence expression at 8- and 16-cell stages. **c** Proximity of RNA (white arrowheads) and LAMP1-3xeGFP positive vesicular structures (orange arrowheads) at 8-cell stage and **d** late 16-cell stage. **e** High spatiotemporal resolution time-series imaging of RNA (grey arrowheads) and LysoSensor-positive vesicular structures (orange arrowheads) during co-trafficking events. **f** At 16-cell stage Annexin A11 (ANXA11-mEmerald) tethers RNA and LAMP1-positive vesicular

structures (blue arrowhead) and **g** is present between LAMP1-positive vesicular-like structures and microtubule filaments (blue arrowheads). **h** Apicobasal RNA distribution in control and Annexin A11 siRNA-microinjected embryos. **i** Developmental rate of control, Annexin A11 siRNA-, Borcs7 siRNA- and Annexin-R346C mutant-microinjected embryos. Data are represented as a percentages (mean); n = embryos. **j** Apicobasal RNA distribution in control, Borcs7 siRNA- and Annexin-R346C mutant-microinjected embryos. Box plots in **b**, **h** and **j** display minimum, lower quartile, median, upper quartile and maximum. Unpaired Mann-Whitney or Welch's two-tailed t-tests were used to identify statistical differences. Scale bars 10 μm; 1 μm insets; except insets in (**a**) 5 μm. Source data are provided as a Source data file.

homogenously dispersed LAMP1-3xeGFP-labelled vesicular structures in all blastomeres (Fig. 3a and Supplementary Fig. 3a). Coincidently with the establishment of apical-to-basal RNA asymmetries at late 16-cell stage, however, LAMP1-3xeGFP-labelled vesicular structures also adopted a predominant basal localisation, demonstrated by a significant increase in LAMP1-3xeGFP fluorescence intensity in basal compared to apical regions (Fig. 3a, b and Supplementary Movie 5). This suggests that the basally directed movement of RNA might be

coupled with the transport of LAMP1-3xeGFP-labelled vesicular structures.

Furthermore, while being distinctly separated at earlier developmental stages (Supplementary Fig. 3a), LAMP1-3xeGFP-labelled vesicular structures and RNA transcripts became more closely positioned to each other as the living embryo progressed through 16-cell stage (Fig. 3c, d). We did not observe a similar spatiotemporal distribution pattern or co-localisation with eGFP-Rab11a or eGFP-G3BP1, markers

for early and recycling endosomes or RNA granules (Supplementary Fig. 3b, c), thus reasoning LAMP1-labelled vesicular structures are most likely lysosomes. In fact, high spatial resolution imaging revealed that RNA was rarely translocated into the lysosomal lumen during pre-implantation embryogenesis (Fig. 3c, d). Instead, at 16-cell stage, RNA was associated with lysosomes leading the direction of transport, as if RNA being pulled along by lysosomes (Fig. 3e and Supplementary Movie 6).

In accordance with studies in other cell systems[25], we identified Annexin A11, fluorescently tagged to mEmerald (ANXA11-mEmerald), as the anchor between LAMP1-positive vesicular structures and RNA by performing imaging of co-labelled 16-cell stage preimplantation mouse embryos (Fig. 3f and Supplementary Fig. 3d). In addition, ANXA11-mEmerald coated LAMP1-positive vesicular structures were positioned along microtubule filaments (Fig. 3g). Knockdown of ANXA11 prevented the establishment of RNA and lysosome asymmetries at late 16-cell stage (Fig. 3h and Supplementary Fig. 3e, f) and delayed the subsequent development of embryos towards 32-cell stage (Fig. 3i and Supplementary Fig. 3g). This delay in embryonic development was reproduced by the overexpression of the ANXA11-R346C mutant[25], only disrupting RNA asymmetries, but not by the downregulation of BLOC-1 Related Complex Subunit 7 (Borcs7)[26] preventing the microtubule plus end-regulated localisation of lysosomes basally (Fig. 3i, j and Supplementary Fig. 3e–h).

Thus, the spatiotemporally-controlled transport of RNA, involved in the timely progression of mammalian preimplantation embryogenesis, requires the tripartite complex of microtubules, lysosomal vesicular structures and Annexin A11.

## Higher translation activity of apical RNA transcripts

To investigate if the asymmetric localisation of global RNA transcripts leads to a preferential subcellular site of protein synthesis, we sought to determine the presence of mRNA, rRNA and tRNA subtypes in basal clusters and apically dispersed foci of global RNA. The subcellular localisation of mRNAs and tRNAs were visualised by performing fluorescence in situ hybridisation (FISH) using fluorescein-tagged oligonucleotides complementary to mRNA poly(A) tails and mammalian tRNA$^{Lys}$[27,28]. While basal regions were mostly deprived of both types of RNA, apical regions of outer blastomeres at late 16-cell stage showed a strong expression of fluorescein-labelled mRNA and tRNA puncta, co-localised with global RNA foci (Fig. 4a–d and Supplementary Fig. 4a–c). Contrary, rRNA was not only present at apical RNA foci, but also in basal regions visualised by using the rRNA-specific live dye, rRNA-naphthalimide[29] (Fig. 4e, f and Supplementary Fig. 4d). Therefore, our data demonstrate an accumulation of all three RNA subtypes required for protein synthesis at the apically dispersed RNA foci, but not in rRNA-enriched basal regions.

To further investigate if the differences in rRNA localisation in basal and apical regions relate to the function of ribosomes, we visualised the expression of the 60S large ribosomal subunit tagged with eGFP (L10A-eGFP)[30] and the 40S small ribosomal subunit tagged with mScarlet (RPS6-mScarlet). L10A-eGFP fluorescence intensity was asymmetrically increased apically in outer blastomeres of late 16-cell stage embryos, while RPS6-mScarlet showed a more homogenous distribution (Fig. 4g). Furthermore, their co-localisation was increased in apical regions, suggesting a higher abundance of assembled ribosomal subunits apically (Fig. 4g and Supplementary Fig. 4e).

The synthesis of ribosomal subunit proteins and rRNA expression is coordinated by the RNA-binding protein HIV-1 Tat Specific Factor 1 (HTATSF1)[31]. HTATSF1 downregulates ribosomal protein synthesis through intron retention but upregulates rRNA expression by removing the internal transcribed spacer 1 (ITS1) sequence from rRNA[31]. Accordingly, live imaging of HTATSF1 fused to mScarlet (HTATSF1-mScarlet) in outer blastomeres of 16-cell stage embryos revealed an enrichment of HTATSF1-mScarlet in basal regions (Fig. 4h and

Supplementary Fig. 4e). This suggests that HTATSF1-dependent ribosomal biogenesis is differentially regulated in apical and basal regions of 16-cell stage outer blastomeres, resulting in adjoint ribosomal subunit proteins and rRNA apically.

Next, we asked if the apically increased presence of mRNA, tRNA and ribosomal complexes may serve as a mechanism to regulate spatially differential gene expression and extended our studies to components of the translational and post-translational machinery. Apical RNA foci were entrapped and surrounded by prominent dense ribosomal regions labelled with L10A-eGFP (Fig. 5a, e). Despite a more dispersed and scattered expression pattern, some basally located L10A-eGFP ribosomal subunits were attached to the larger RNA clusters (Fig. 5a). Evident apicobasal heterogeneities were also observed for the endoplasmic reticulum (ER; Emerald-Sec61β) (Fig. 5b, e). At apical regions, a tubular-like ER network was closely associated with RNA foci compared to some sporadic less defined clustered basal ER fragments (Fig. 5b). We next investigated the localisation pattern of two translation initiators, eukaryotic translation initiation factor 2, subunit β (eIF2β) and Polyadenylate-Binding Protein Cytoplasmic 1 (PABPC1). The real-time fluorescence expression of eGFP-eIF2β, combined with RNA, matched the asymmetric apical expression pattern of ribosomes and ER (Fig. 5c, e). Conversely, PABPC1-eGFP, a protein which stabilises newly synthesised 3′ ends of mRNAs[32], did not surround but overlapped with the majority of apical but only very few basal RNA foci (Fig. 5d, e).

Altogether, these results indicate that the apical region with more dispersed RNA is the translationally more active site compared to the basal site of outer 16-cell stage blastomeres.

To determine if apical RNA foci undergo active translation, we adopted the ribopuromycylation method (RPM)[33–35] in late 16-cell stage preimplantation mouse embryos. In line with our results of RNA subtypes (Fig. 4a–f) and translation components (Fig. 5a–e), ribosome-attached puromycylated nascent peptides were predominantly found at apical L10A foci (Fig. 6a, b and Supplementary Fig. 4f, g). 29.4 ± 0.014% of apical L10A foci co-localised with puromycin, indicative of actively translating ribosomes, compared to 11.1 ± 0.021% in basal regions. Instead, 69.2 ± 0.038% of basal foci were labelled only for L10A but not puromycin, indicative of a high number of translationally quiescent ribosomal subunits (Fig. 6a, b and Supplementary Fig. 4g). Furthermore, apical RNA foci co-localised more frequently with puromycin (31.6 ± 0.024%) compared to basal RNA clusters (24.56 ± 0.033%) (Fig. 6c, d and Supplementary Fig. 4h), while majority of basal RNA foci were Puromycin-negative (67.9 ± 0.047%) (Fig. 6c, d and Supplementary Fig. 4h).

To further confirm the engagement of F22-labelled apical global RNA foci in translation events, we used RNA dynamics as a measurement of protein synthesis as ribosome-associated mRNAs undergoing translation exhibit slower and more spatially confined movements[30,36]. In control conditions, tracking of RNA foci speed in apical and basal regions of late 16-cell stage outer blastomeres did not reveal any significant differences (Fig. 6e). A short 5 min treatment with the translation elongation inhibitor cycloheximide caused a slight reduction in RNA foci speed from 0.21 ± 0.04 μm/s to 0.186 ± 0.053 μm/s in apical regions and from 0.180 ± 0.012 μm/s to 0.144 ± 0.026 μm/s basally (Fig. 6e). After a 2 h cycloheximide treatment, however, apical RNA foci accelerated to 0.924 ± 0.120 μm/s and basal RNA to 0.276 ± 0.020 μm/s. This increase in speed was confirmed by using the alternative translation inhibitor puromycin (Fig. 6e). Collectively, these findings suggest that components required for translation display a preferential apical localisation in outer blastomeres of 16-cell stage embryos, coupled with increased translation capacity.

To visualise translation events in real-time and to track the spatiotemporal dynamics of translation in 16-cell stage blastomeres of the mouse preimplantation embryo, we employed the SunTag system for

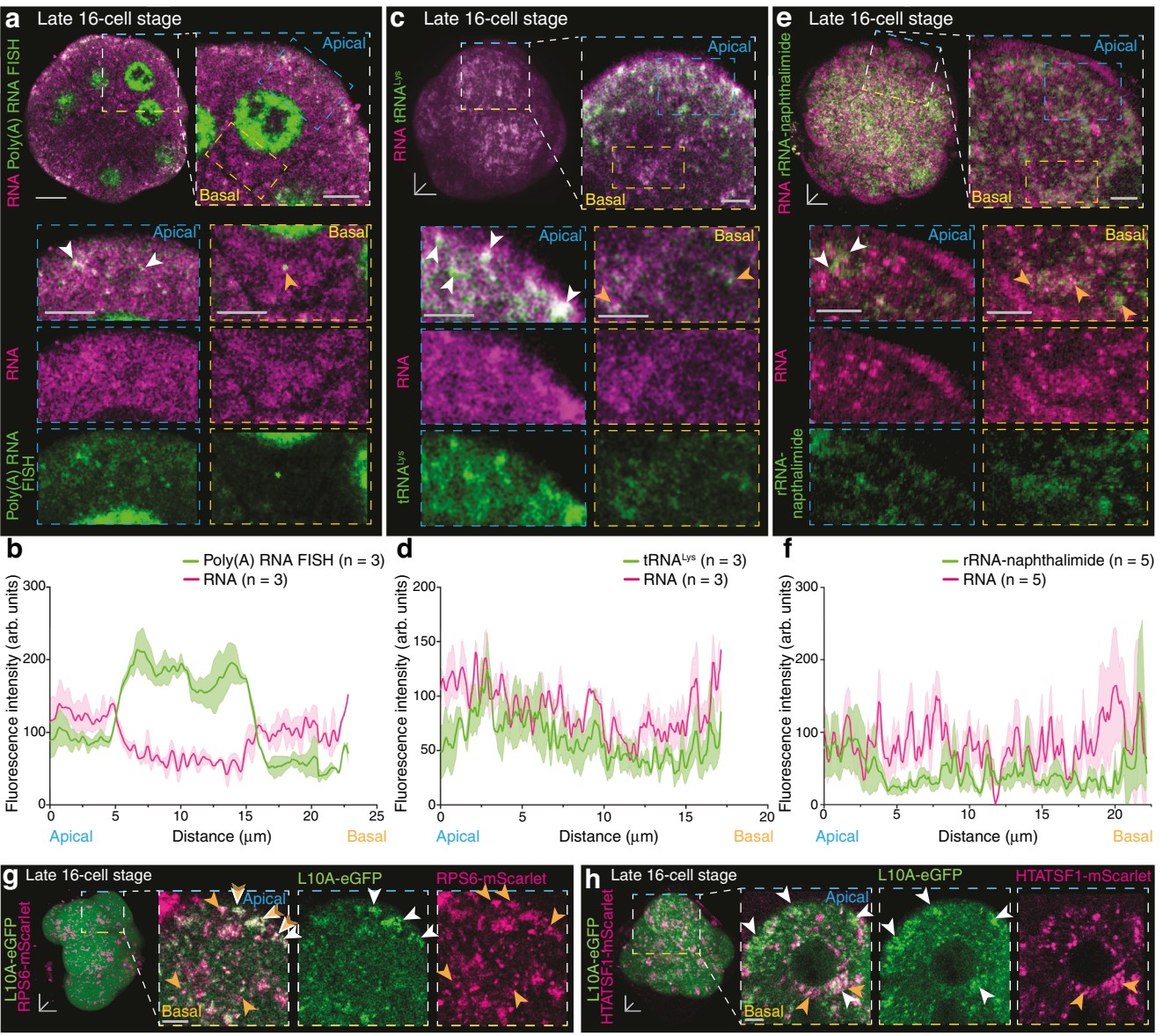

**Fig. 4 | RNA subtypes required for protein synthesis are enriched apically in late 16-cell stage outer blastomeres.** Late 16-cell stage embryos show an apical enrichment (white arrowheads) for **a**, **b** mRNA (Poly(A) RNA FISH) and **c**, **d** tRNA FISH. **e**, **f** Global RNA and ribosomal RNA (rRNA-naphthalimide) co-clusters in apical (white arrowheads) and basal (orange arrowheads) regions of late 16-cell stage blastomeres. **g** Small 40S (RPS6-mScarlet) and large 60S ribosomal subunit (L10A-eGFP) expression in apical (white arrowheads) and basal (orange arrowheads) regions. **h** Increased basal RNA-binding protein HTATSF1 (HTATSF1-mScarlet) (orange arrowheads) distribution with L10A-eGFP co-labelling (white arrowheads) in 16-cell stage outer blastomere. Data in **b**, **d** and **f** are presented as mean ± SEM; thick lines indicate mean, transparent shadows depict SEM. Scale bars 10 µm. Source data are provided as a Source data file.

the translation initiation factor eIF2β[37,38] (Fig. 6f–h and Supplementary Fig. 4i–p). In line with our observation of increased eGFP-eIF2β protein expression apically (Fig. 5c), scFv-sfGFP-labelled eIF2β proteins were more abundant in apical regions (Fig. 6f, g, Supplementary Movie 7 and Supplementary Fig. 4i, j). 50.92 ± 3.64% of scFv-sfGFP-labelled eIF2β proteins were co-localised with PCP-2xmCherry-coated *eIF2b* mRNAs, indicative of translation events, compared to 31.04 ± 5.349% in basal regions (Supplementary Fig. 4j). Such translation hotspots were more prominent in apical regions compared to basal regions (Fig. 6f, g and Supplementary Fig. 4i, j) and could be tracked for approximately 2.5 min (Fig. 6h and Supplementary Fig. 4k). Thus, the localisation of *eIF2b* mRNA in apical regions, as confirmed by *eIF2b* RNA FISH (Supplementary Fig. 4l–n), leads to a subcellular site-specific protein synthesis of eIF2β, specifically in apical areas of increased translation activity.

## Asymmetric inheritance of RNA and translation components determines cell fate

We next asked what the consequence of the asymmetries in apical-to-basal RNA and translation is on cell fate and developmental potential of the embryo. We first tracked dividing 16- to 32-cell stage pre-implantation mouse embryos to determine if the asymmetries are maintained beyond 16-cell stage and might bias the generation of cells with differential fate. We found that following symmetric cell divisions, where the division plane was located perpendicular to the basal membrane, RNA was evenly inherited between sister blastomeres (Fig. 7a, Supplementary Fig. 5a and Supplementary Movie 8). For each symmetric division, the mother and both daughter blastomeres maintained a constant RNA fluorescence intensity, normalised to the cell volume (Fig. 7b), which remained consistent throughout the 32-cell stage (Fig. 7b). 6.5 h after division, both sister cells remained at the

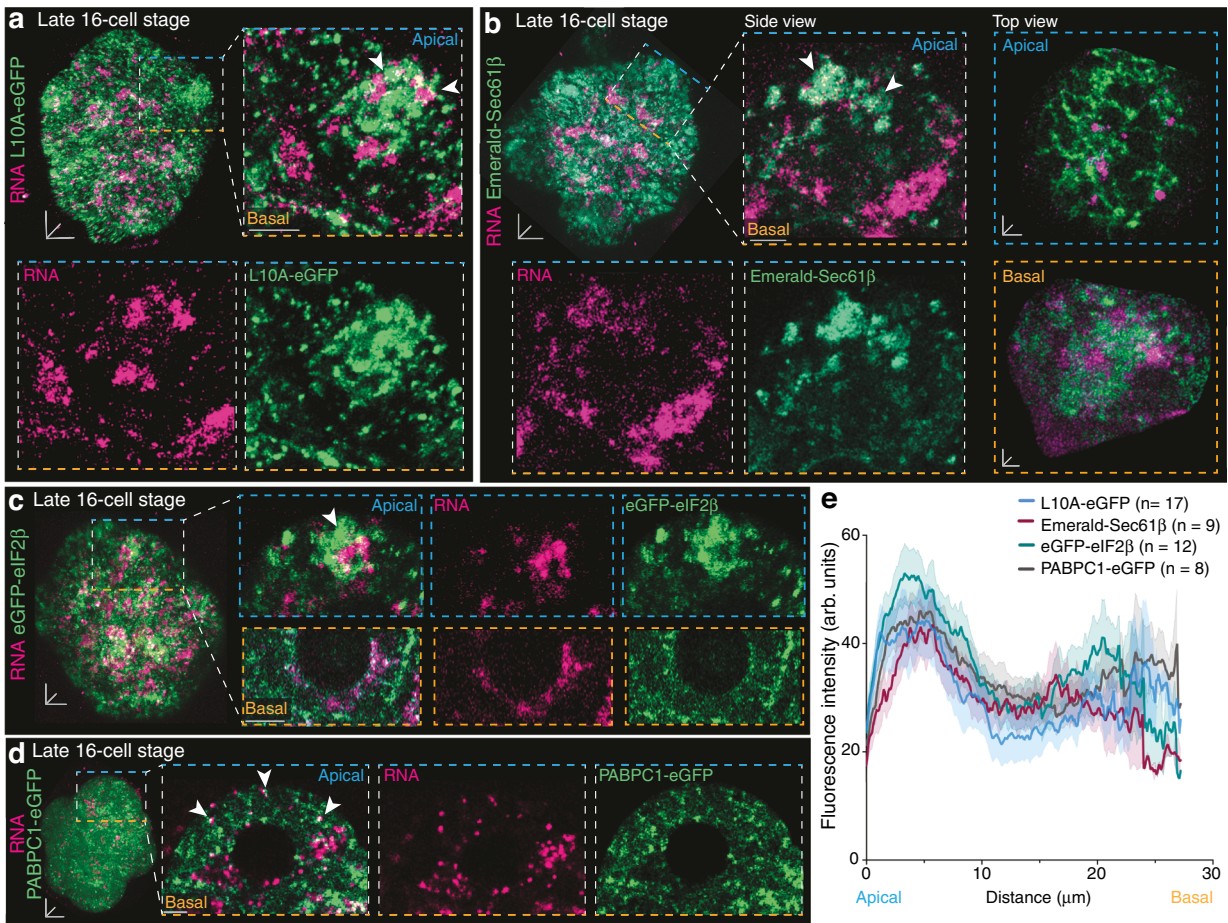

**Fig. 5 | Apical enrichment of translation components. a–d** Late 16-cell stage embryos show apical enrichment (white arrowheads) for **a** large 60S ribosomal subunit L10A-eGFP, **b** Endoplasmic reticulum (Emerald-Sec61β) (side view) with an apical tubular-like network compared to ER in basal regions (top view), **c** eukaryotic translation initiation factor 2β (eGFP-eIF2β) and **d** Polyadenylate-Binding Protein Cytoplasmic 1 (PABPC1-eGFP), with RNA foci. **e** Fluorescence intensity quantification of 16-cell stage outer blastomeres from apical to basal regions for (**a–d**). Data in **e** presented as mean ± SEM; thick lines indicate mean, transparent shadows depict SEM. Scale bars 10 μm; insets 4 μm; except top and side view insets in b 2 μm. Source data are provided as a Source data file.

outside of the embryo (Fig. 7c) and expressed trophectoderm marker CDX2 (Fig. 7d).

In contrast, during asymmetric divisions occurring parallel to the basal membrane, RNA was non-uniformly partitioned (Fig. 7e, Supplementary Fig. 5b and Supplementary Movie 9). Initially, mother blastomeres dividing asymmetrically showed a reduced normalised fluorescence intensity in total RNA compared to symmetrically dividing mother blastomeres (2.11 ± 0.06 arbitrary units (arb. units) and 3.5 ± 0.38 arb. units, respectively) (Fig. 7f). Furthermore, basally located RNA clusters of late 16-cell stage mother blastomeres were predominantly inherited by the internalising daughter blastomere and thus the normalised RNA fluorescence intensity increased to 2.52 ± 0.72 arb. units (Fig. 7e–g). Consequently, the blastomere inheriting the lesser apical RNA foci gave rise to an outer blastomere and assumed a reduced normalised RNA fluorescence intensity of 1.75 ± 0.45 arb. units (Fig. 7e–g). Performing correlative immunostainings of the live tracked embryos at late 32-cell stage (~8 h after division) revealed the expression of CDX2 in outer blastomeres, inheriting less RNA, but not in inner RNA-enriched blastomeres (Fig. 7h). This suggests that RNA asymmetries at the late 16-cell stage are required for cell fate allocation of inner and outer blastomeres at 32-cell stage.

As the amount of RNA varied between outer blastomeres deriving from a symmetric or asymmetric division, we questioned whether they also inherited the apically, more translationally active RNA

components. Live tracking of asymmetric cell divisions of 16- to 32-cell stage embryos labelled for L10A-eGFP demonstrated that the initially apically enriched ribosomes dispersed along the cytokinetic furrow, resulting in a more even inheritance of ribosomes between inner and outer 32-cell stage blastomeres (Fig. 8a, b). Tracking asymmetric cell divisions in embryos labelled for Emerald-Sec61β and BFP-Utrophin, however, revealed that the ER was asymmetrically inherited, resulting in an increased ER intensity in the blastomeres destined for outer cell fate (Fig. 8c, d and Supplementary Movie 10). We confirmed an increased ER density in 32-cell stage outer blastomeres compared to inner blastomeres using serial blockface scanning electron microscopy (SBF-SEM) as well as immunostainings (Figs. 8e, f and 9a). Moreover, we uncovered fibril-like ER structures in outer blastomeres which were significantly longer (0.83 ± 0.04 μm in outer cells; 0.39 ± 0.02 μm in inner cells) and preferentially surrounding cytoplasmic lattice structures, known to harbor ribosomes in preimplantation embryos[17,39] (Fig. 8e, f). In inner blastomeres, however, ER-associated cytoplasmic lattices were rare (Fig. 8e).

To uncover if our findings of differential global RNA (Fig. 7e–g) and ER inheritance (Fig. 8c, d) are associated with cell fate determination, we compared SEC61β expression levels to CDX2 and SOX2 to identify trophectoderm (CDX2-positive/SOX2-negative), epiblast (CDX2-negative/SOX2-positive) and primitive endoderm (CDX2-negative/SOX2-negative) progenitors. CDX2-positive/SOX2-negative outer cells had the highest SEC61β fluorescence expression intensity

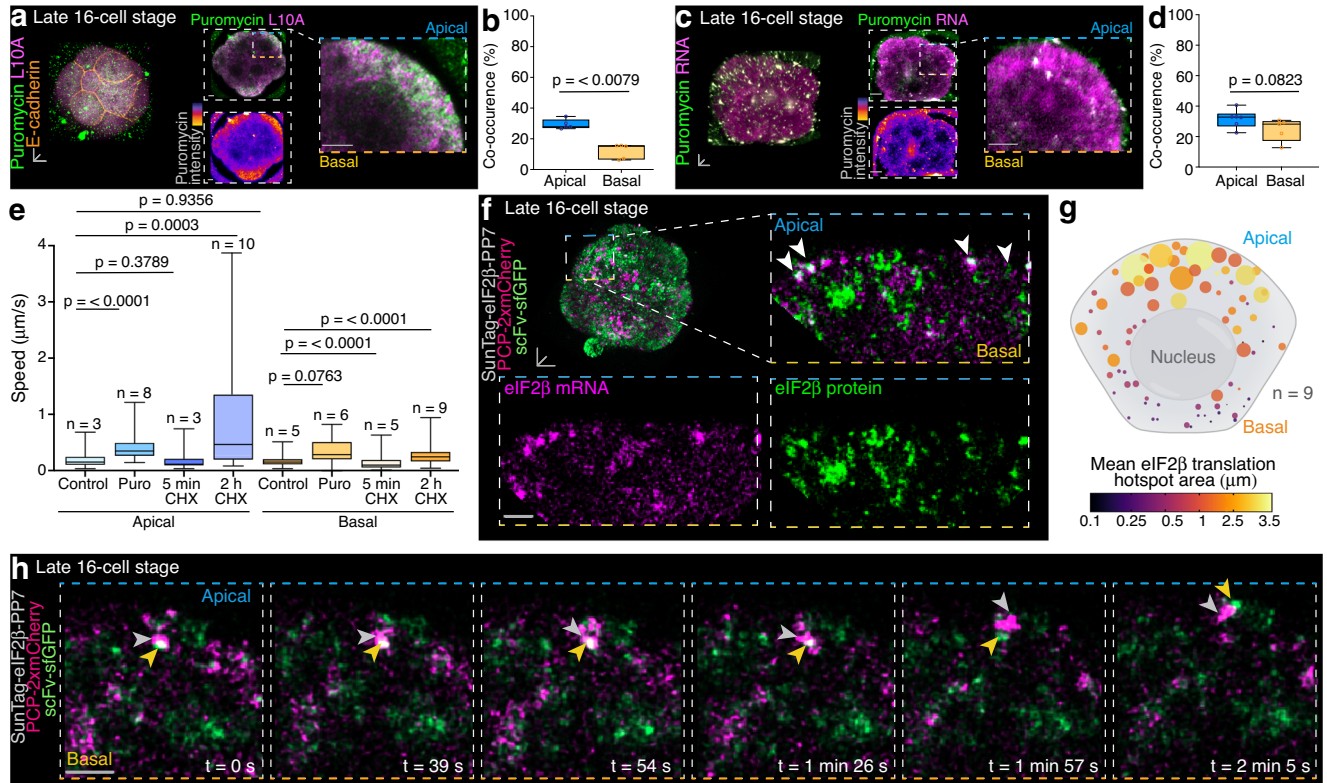

**Fig. 6 | Higher translation activity at apical regions of late 16-cell stage outer blastomeres. a** Ribopuromycylation method (RPM) in late 16-cell stage embryos, with **b** quantification of co-occurrence percentage of L10A (ribosomes) and Puromycin in apical and basal regions. **c** Puromycylated nascent RNA transcripts co-localise with global RNA foci, **d** quantified in apical and basal regions. **e** RNA foci speeds following puromycin (Puro) and cycloheximide (CHX) treatment in apical and basal regions. **f** Late 16-cell stage embryo expressing the translation reporter (SunTag-eIF2β-PP7) combined with sfGFP-tagged single-chain variable fragment antibody (scFv-sfGFP) and PP7-bacteriophage coat protein (PCP-2xmCherry) enables the visualisation of translating eIF2β mRNA foci. **g** Quantification of translation hotspot number in apical and basal regions. **h** Timeseries of translating eIF2β mRNA hotspot proximal to the apical membrane. Box plots in **b**, **d** and **e** display minimum, lower quartile, median, upper quartile and maximum. Unpaired Mann-Whitney two-tailed t-tests were used to identify statistical differences. Scale bars 10 μm; insets 4 μm; except insets in (**f**) and (**h**) 0.5 μm. Source data are provided as a Source data file.

(77.41 ± 2.5 arb. units) (Fig. 9a, b). Moreover, we observed significant differences in SEC61β fluorescence expression intensity for CDX2-negative/SOX2-positive epiblast progenitor cells (52.41 ± 3.66 arb. units) and CDX2-negative/SOX2-negative primitive endoderm progenitor cells (35.17 ± 3.6 arb. units) (Fig. 9a, b).

Next, we explored if the first cell fate decision during pre-implantation embryogenesis is regulated by the inheritance of sub-cellular asymmetries established at 16-cell stage. First, we asked if the inherited RNA at 32-cell stage contained transcriptional information to mediate cell fate decisions. Therefore, we selected *Cdx2/Sox2*-controlled cell fate regulators from available ChIP-seq data sets[40] (Supplementary Fig. 5c–f) in embryonic stem cells (as a surrogate of epiblast) and embryonic stem cells derived trophoblast stem cell-like cells (as a surrogate for trophectoderm). We selected the Transcription factor AP-2γ (*Tfap2c*), which mediates trophectoderm development[41–43], and for which *Cdx2* binding was observed in trophoblast stem cell-like cells. We also selected Nanog, which is considered as a core epiblast transcription factor, and is a target of *Sox2*[44]. Using RNA FISH, we visualised the spatial localisation of *Tfap2c* and Nanog mRNA transcripts in blastomeres of 32-cell stage embryos. In addition to the expected nuclear expression, both *Tfap2c* and Nanog mRNA transcripts co-localised with some global RNA foci in the cytoplasm of outer and inner 32-cell stage blastomeres respectively (Supplementary Fig. 5g, h).

Secondly, we uncoupled RNA from the lysosomes by the down-regulation of ANXA11 to disable their basally-directed transport along microtubule filaments. We immunostained ANXA11 siRNA-microinjected embryos, displaying a lack of RNA and lysosome asymmetries from 16- to 32-cell stage (Fig. 3h and Supplementary Fig. 3e, f), with the cell lineage markers CDX2 and SOX2. Contrary to the nuclear mosaic expression pattern of control embryos, SOX2 was detected in all blastomeres of ANXA11 siRNA-injected embryos in both cytoplasmic (34.26 ± 1.13 arb. units) and nuclear compartments (39.66 ± 2.18 arb. units) at similar expression levels (Fig. 9c, d and Supplementary Fig. 5i). Furthermore, Cdx2 was not only expressed in outer blastomeres (66.91 ± 3.39 arb. units) but also, to weaker extent, in some inner blastomeres of ANXA11 siRNA-injected embryos (36.49 ± 4.667 arb. units) (Fig. 9c and Supplementary Fig. 5j). A reduction in asymmetric expression between outer and inner blastomeres was also observed for the ER (Supplementary Fig. 5k, l). The inability of ANXA11 siRNA-injected embryos to specify the cell fate of inner and outer blastomeres was accompanied by a significant reduction of inner blastomeres (4.45 ± 0.47) compared to 10 ± 1.7 in control embryos (Fig. 9e). Accordingly, the total blastomere number of ANXA11 siRNA-injected embryos was reduced, from 29.83 ± 1.49 in control embryos, to 24 ± 1.16 (Fig. 9e).

These results demonstrate that apicobasal asymmetries established at the 16-cell stage bias the inheritance of RNA and translation components to outer and inner 32-cell stage blastomeres. Yet, the decisive factor for cell fate allocation is not only the total amount of RNA in each blastomere but the translational capacity of the inherited RNA pool (Fig. 10).

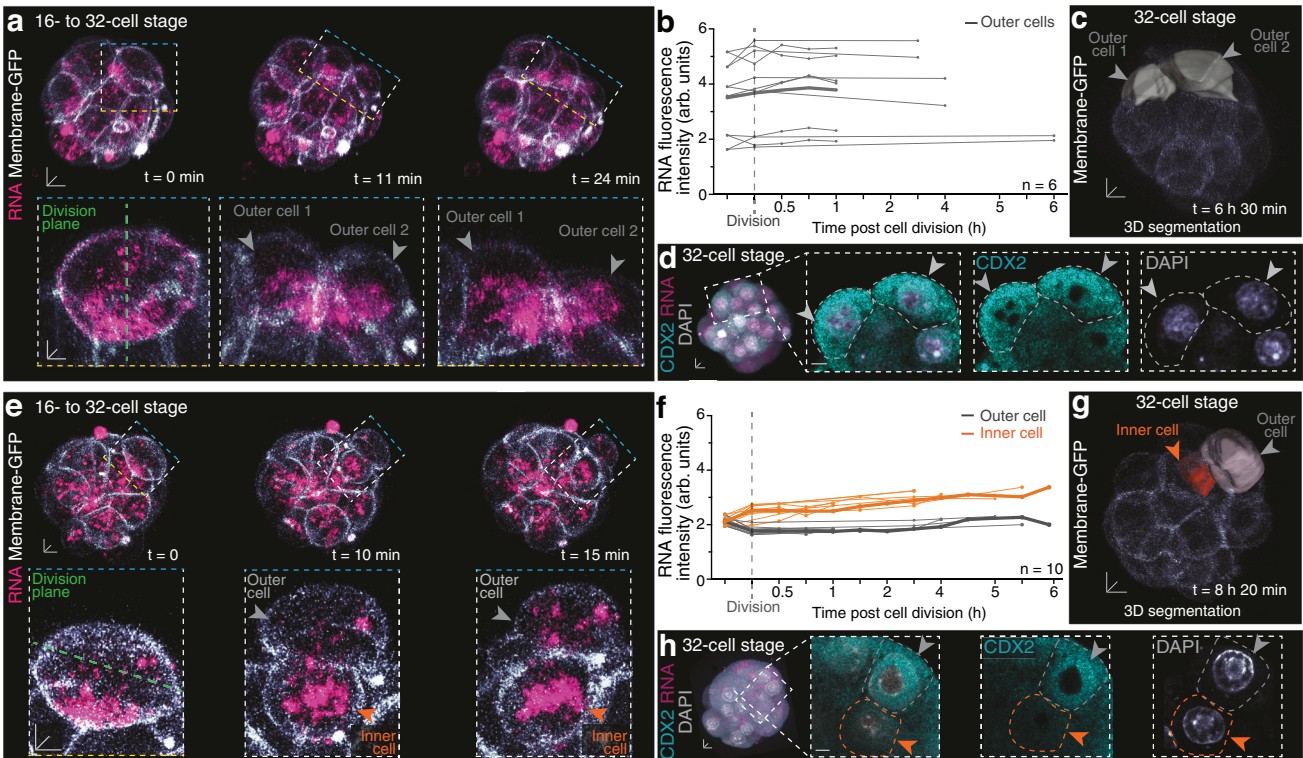

**Fig. 7 | Biased inheritance of RNA following asymmetric cell divisions from 16- to 32-cell stage. a** Symmetric cell division of dividing 16- to 32-cell stage embryo labelled for RNA and Membrane-GFP. **b** RNA intensity profiles from 16-cell stage mother cells (pre-division) to outer 32-cell stage daughter cells during symmetric cell divisions (dashed vertical line); bold line indicates mean of individual tracked divisions (faint grey lines). **c** Segmentation of tracked blastomeres (grey) in (**a**), ~6.5 h following symmetric cell divisions and **d** correlative immunostaining of CDX2 with close-up view of outer daughter cells (grey arrowheads). **e** Asymmetric cell division of dividing 16- to 32-cell stage embryo labelled for RNA and Membrane-GFP. **f** RNA intensity profiles of inner sister (orange lines) and outer sister blastomeres (faint grey lines) following asymmetric cell divisions; bold line indicates mean values. **g** Segmentation of tracked blastomeres in **e**, 8 h 20 min following asymmetric cell division demonstrates outer (grey) and inner (orange) positions and **h** correlative immunostaining of CDX2 with close-up view of outer (grey arrowhead) and inner (orange arrowhead) daughter cell. Scale bars 10 µm; insets 5 µm. Source data are provided as a Source data file.

## Discussion

Since its first discovery in ascidian embryos in 1983[45], the list of specific asymmetrically inherited RNA transcripts has expanded to encompass an estimated 71% of RNAs during *D. melanogaster* embryogenesis[46]. Despite its importance, the nature and extent of RNA asymmetries in mammalian embryos has remained elusive. To identify the global prominence of RNA asymmetries during mammalian preimplantation development, we performed live imaging using a fluorescent RNA dye, enabling us to visualise the direct interplay of RNA with the cytoskeleton and organelles. Intriguingly, we discovered subcellular heterogeneities of RNA subtypes and translation events during the second wave of asymmetric cell divisions in mammalian preimplantation embryos. At this time, the first cell lineages are specified by the position of the blastomeres, cell polarity and the differential expression of signalling and transcription factors, for instance *Sox2*, *Cdx2*, Notch or Hippo[1,2,6]. Our results using live imaging of RNA transcripts in their spatial and temporal context provides insights into how molecular dynamics act synergistically with the cellular organisation and gene expression patterns of cells of the preimplantation embryo to regulate cell fate decisions.

Asymmetric segregation of fate determinants in stem cells, including RNA and proteins, is typically ensured by the orientation of the mitotic spindle, made up of microtubule filaments[47]. Here, we uncover a preceding role on how the microtubule cytoskeleton establishes subcellular asymmetries during interphase. As the embryo progresses through 16-cell stage, RNA movement is guided by an asymmetric and polarised microtubule network[9,48] toward the basal membrane. In addition to the long-distance transport along

microtubules[25], we do not exclude the possibility that other cytoskeletal factors, such as actin, may also contribute to shorter range, particularly cortical, RNA mobility[49]. Given that asymmetric apicobasal targeting of RNA is one of the most conserved patterns across various cellular contexts[50], subcellular localisation of RNAs provides a precise control over targeted protein synthesis[51], organelle interactions[52], morphogen gradients[13] and cell fate segregation[19,47].

In the preimplantation mouse embryo, at late 16-cell stage, basal RNA clusters are spatially and functionally segregated from smaller apical RNA foci. The latter can contain rRNA, mRNA and tRNA, associated with translation components providing a mode of spatial translation regulation[53]. The increased need for translation capacity in apical areas, inherited to 32-cell stage outer blastomeres, might be required for a rapid change in gene expression to promote trophectoderm differentiation[54]. Notably, tubular ER, also inherited from apical areas to outer blastomeres, functions more efficiently by trafficking larger organelles and by mediating microtubule interactions[55]. As such, biased ER inheritance could facilitate cell differentiation, suggesting that the ER serves as an organelle-based marker for cell fate, complementing conventional genetic factors.

Conversely, pluripotency maintenance and function, embodied by the inner cells of the embryo, requires low translation rates[31] and instead the continued transcription of rRNAs[56] accompanied by altered ribosome biogenesis[57,58]. Accordingly, the vast majority of RNAs in the cells of the mammalian preimplantation embryo are rRNAs[39], supporting our finding that a large portion of the densely clustered RNA detected basally is rRNA. Thus, the compartmentalisation of RNA translation mechanisms and rRNA synthesis is emerging as

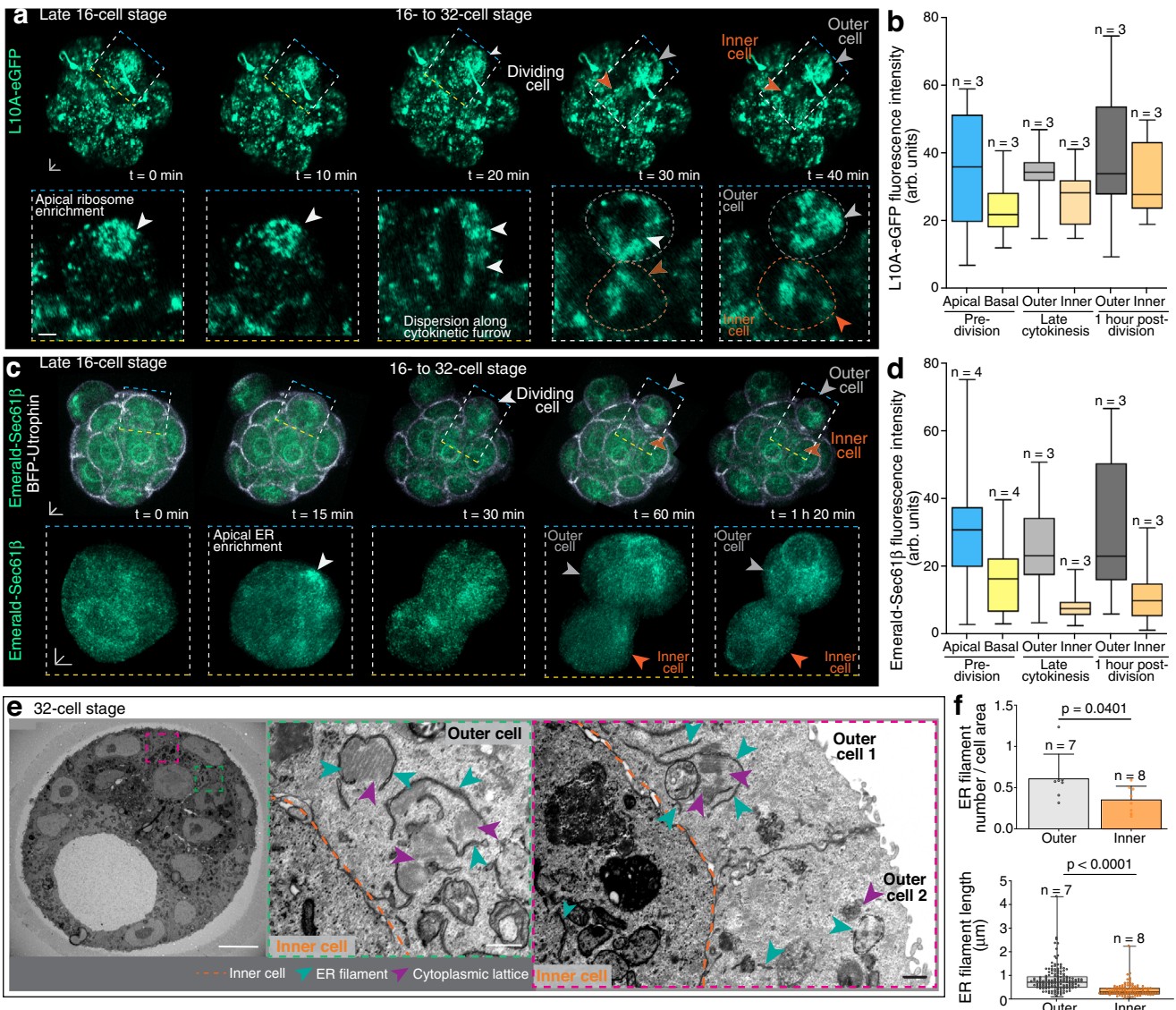

**Fig. 8 | ER is preferentially inherited by and enriched in outer cells at 32-cell stage. a** Partial redistribution of ribosomes (L10A-eGFP) during 16- to 32-cell stage asymmetric divisions. **b** Quantification of ribosome inheritance amongst inner and outer sister blastomeres. **c** Biased inheritance of endoplasmic reticulum (Emerald-Sec61β) following 16- to 32-cell stage asymmetric divisions. Prospective outer blastomere (grey arrowhead) displays higher ER intensity (insets). **d** Quantification of (**c**). **e** Serial blockface scanning electron microscopy (SBF-SEM) images of 32-cell stage embryo show ER filament-like structures (red dashed inset) surrounding cytoplasmic lattice structures in outer blastomeres (green dashed inset). **f** Quantification of ER filaments from SBF-SEM images. Box plots in **b**, **d** and **f** display minimum, lower quartile, median, upper quartile and maximum. Unpaired Mann-Whitney two-tailed t-tests were used to identify statistical differences. Individual data points are overlaid in (**f**). Scale bars 10 μm; insets 5 μm. SBF-SEM scale bars 0.5 μm. Source data are provided as a Source data file.

a determining factor of pluripotency, infertility and reproductive ageing in humans[59–61]. Overall, these findings demonstrate that mRNAs are preferentially translated at apical regions, whereas rRNAs are retained basally to underpin the segregation of trophectoderm cell lineage and inner cells, respectively. Alternatively, RNA enriched at the basal membrane could bias the inheritance of selected mRNA transcripts or small RNA species representative of a pluripotent state, which remains challenging to be visualised in real-time[16].

Downregulation of ANXA11, the anchor between microtubules and RNA-lysosome cargos, prevented the establishment of such apicobasal asymmetries in cellular architecture. As a result, preimplantation development stalled during 16- to 32-cell stage divisions, with striking similarities to *Sox2* siRNA-treated embryos[62]. Contrary to an overall loss of *Sox2* expression and subsequent reduction of outer trophectoderm cells[62], ANXA11-siRNA treated embryos showed a reduction of inner cells with dominant nuclear SOX2 expression, thus

presumably lacking epiblast progenitor cells. Accordingly, an even nuclear and cytoplasmic SOX2 expression and majority of blastomeres being CDX2 positive indicates a primitive endoderm or trophectoderm fate. An absence of epiblast cells was also previously demonstrated in mice genetically downregulated for zygotic but not maternal *Sox2* expression[63]. Thus, in agreement with our findings that apicobasal RNA localisation is correlated with translation capacity, distinct expression levels of SOX2 or other lineage markers[64], bias the specification of cell lineages in the preimplantation embryo.

Deciphering the spatiotemporal RNA localisation code in mammalian embryos and pluripotent cells may facilitate the development of invaluable methods to predict a cell's future transcriptome and fate[65] before the change in fate occurs. Furthermore, coupling spatial transcriptomics[66] with live imaging approaches will provide essential insights into the interplay of the morphological and genetic mechanisms controlling cell fate decisions, a current bottleneck in the stem

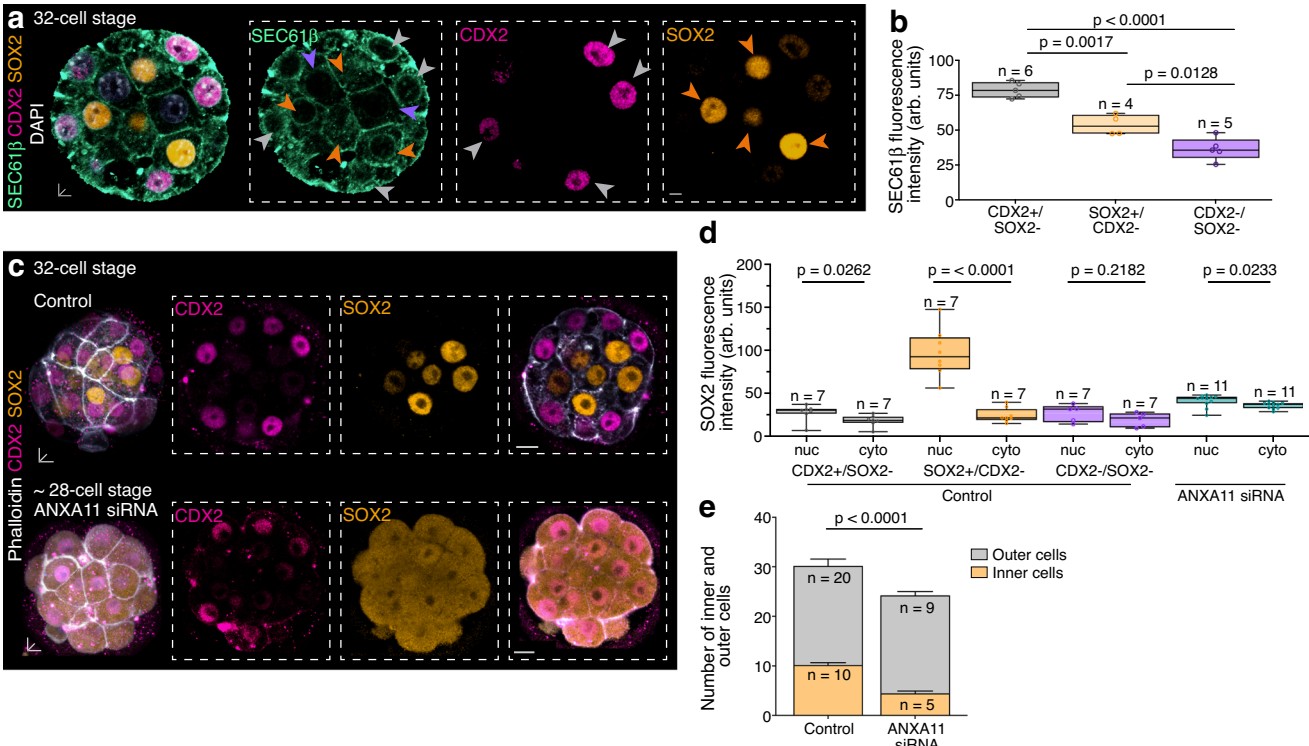

**Fig. 9 | Pluripotent inner cells require lower translation capacity compared to differentiating outer trophectoderm cells. a** Immunofluorescence shows blastomeres positive for trophectoderm marker CDX2 (grey arrowheads) with higher SEC61β intensity compared to SOX2 positive blastomeres (orange arrowheads) and CDX2/SOX2 negative blastomeres (purple arrowheads). **b** Quantification of SEC61β fluorescence intensity. **c** SOX2 and CDX2 expression in control and Annexin siRNA-treated embryos. **d** Quantification of nuclear (nuc) and cytoplasmic (cyto) SOX2 fluorescence intensity. **e** Inner (orange) and outer (grey) blastomere numbers in control and Annexin A11 siRNA-treated embryos after 5th cell division. n = embryos. Box plots in **b**, **d** and **e** display minimum, lower quartile, median, upper quartile and maximum. Unpaired Mann–Whitney or Welch's two-tailed t-tests were used to identify statistical differences. Scale bars 10 μm; insets 5 μm. Source data are provided as a Source data file.

cell field[67]. Such knowledge may enable the delivery of RNA-based therapies with subcellular precision with wide-range applications for regenerative and assisted reproductive medicine.

## Methods

### Mouse embryo work

Experiments were approved by the Monash Animal Ethics Committee (MARP-3) under animal ethics number 19143. Animals were housed in pathogen-free animal house conditions at 23 °C in a 12 h light/dark cycle at the animal facility (Monash Animal Research Platform). Eight- to nine-week-old FVB/N wild-type female mice were superovulated using 5 iu of pregnant mare serum gonadotrophin (Prospec Bio, HOR-272) intraperitoneally, followed 48 h later by 5 iu of human chorionic growth hormone (Sigma-Aldrich, C8554) and mated with FVB/N stud males[9,48], in strict accordance with ethical guidelines of Monash University.

Embryos were flushed from oviducts of plugged females using M2 medium (Sigma, M7167) and cultured in KSOM (Cosmo-Bio, CSR-R-B074) covered with mineral oil (Sigma, M5904), at 37 °C and 5% $CO_2$. Isolated live 1-cell stage embryos were microinjected with in vitro transcribed RNA.

### Plasmids and cloning

Cloning procedures were carried out using NEBuilder® HiFi DNA Assembly Master Mix (New England BioLabs, E2621L). In brief, the fragment of interest was amplified by PCR using Q5® High-Fidelity DNA Polymerase (New England BioLabs, M049S) and was cloned into the pCS2⁺ donor vector which contains a SP6 site for RNA production, with the exception of pCS2⁺ LAMP1-3xeGFP. The mScarlet-1 fluorescent protein from pCS2⁺ LAMP1-mScarlet (this paper) was replaced with

3xeGFP from pHAGE-IRES-puro-NLS-dPguCas13b-3xEGFP-NLS-3xFlag (Addgene #132400).

To clone the SP6-24xGCN4_v4-Kif18b-24xPP7 plasmid the SP6 site was inserted into the original pcDNA4TO plasmid and referred to as SunTag-Kif18b-PP7 in this paper. For cloning of SP6-24xGCN4_v4-eIF2β−24xPP7 the previously cloned SP6-24xGCN4_v4-Kif18b-24xPP7 plasmid was digested with AgeI and NheI-HF to remove the Kif18b sequence, which was then replaced with the eIF2β fragment was amplified by PCR. The SP6-24xGCN4_v4-eIF2β−24xPP7 was referred to as SunTag- eIF2β-PP7 in this paper.

Primers including the insertion of the SP6 promoter were used to amplify the eGFP-N1mEIF2βWT fragment from peGFP-N1meIF2βWT (Addgene #45902). PCR products were verified by PCR gel electrophoresis and purified using PCR purification kit (Sangon Biotech, B610363). All primers and plasmids used in this study are described in Tables S1 and S2.

### mRNA preparation

In vitro synthesis of RNA was performed using the mMessage mMachine High Yield Capped RNA Transcription Kit (Ambion, AM1340) to produce SP6-promotor driven RNA from linearised plasmid DNA templates following the manufacturer's instructions. RNA was purified using the RNA purification kit (Sangon Biotech, B511361-0100). RNAs were diluted in injection buffer (5 mM Tris, 5 mM NaCl, 0.1 mM EDTA, pH 7.4) and microinjected into presumptive zygotes as follows: Membrane-GFP, LAMP1-3xeGFP, LAMP1-mScarlet, L10A-eGFP, Emerald-Sec61β, eGFP-eIF2β, PABPC1-eGFP, ANXA11-mEmerald, mScarlet-Borcs7, RPS6-mScarlet, HTATSF1-mScarlet, 24xGCN4_v4-Cdx2-24xPP7, 24xGCN4_v4-eIF2β−24xPP7, ScFv-sfGFP at 35 ng; eGFP-

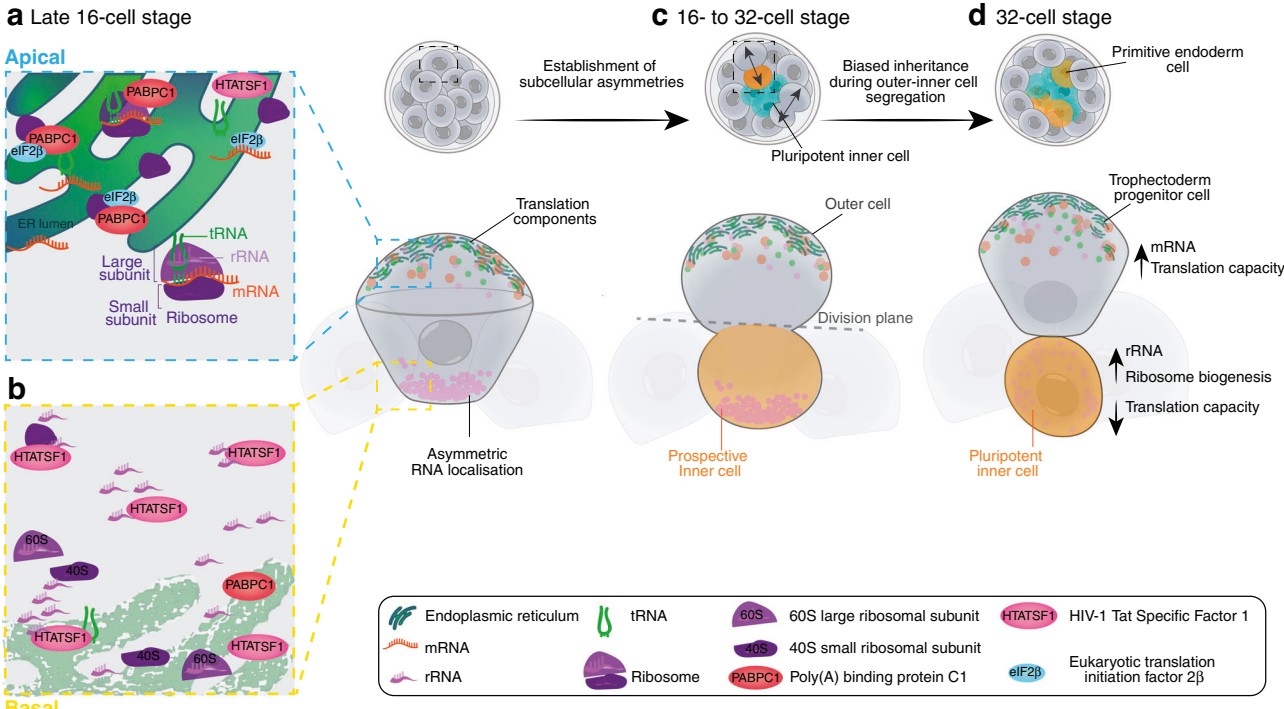

**Fig. 10 | Schematic representation of subcellular heterogeneities in the pre-implantation mouse embryo. a**, **b** Late 16-cell stage embryos establish differentially equipped apical and basal domains to compartmentalise translation activities in outer blastomeres. **c** During asymmetric cell divisions from 16- to 32-cell stage, translation components localised apically are inherited by outer daughter cells, whereas the basal rRNA-enriched RNA pool is inherited by inner daughter cells. **d** The inheritance of such subcellular RNA heterogeneities lead to a higher translation capacity in outer cells which may promote trophectoderm differentiation. Less translation capacity is required by pluripotent inner cells, while maintaining a high rRNA content.

MAP2c, G3PB1-eGFP, mTFP-Utrophin, BFP-Utrophin, PCP-2xmCherry, eGFP-Rab11a at 30 ng. ANXA11-R346C-mEmerald was microinjected at 90 ng. siRNAs (Qiagen) were microinjected at 400 nM described in Table S1.

## RNA riboprobe generation and synthesis

Riboprobes were generated using a modified protocol adapted from ref. 68. DNA template for riboprobe synthesis was achieved by Q5 PCR amplification, primers used included the insertion of the SP6 promoter. PCR products were verified by PCR gel electrophoresis and DNA extracted using DNA Gel extraction kit (Sangon Biotech, B610363). Purified PCR plasmid DNA generated was used as a template for SP6-promotor driven in vitro synthesis reaction. For a 20 µl reaction reagents were added in the following order; 500 ng of plasmid DNA, RNase-free water (up to 20 µl), 2 µl of 10x transcription buffer (Roche, 10999644001), 2 µl of Fluorescein RNA Labeling Mix (Roche, 11685619910), 1 µl of RNaseOUT (Invitrogen, 10777019), SP6 polymerase (Thermo Scientific, EP0131). The reaction was incubated for 2 h at 37 °C, subsequently 2 µl of DNase was added and incubated for an additional 15 min. The reaction was stopped with 1 µl of 0.2 M EDTA pH 8. Subsequently, riboprobe was purified via RNA precipitation by adding the following: 30 µl of RNase-free water, 30 µl of 3 M sodium acetate pH 5.2 and 125 µl of RNase free 100% ethanol, mixed well and kept at −20 °C for at least 20 min. The reaction was centrifuged at maximum speed for 20 min at 4 °C, supernatant discarded, and pellet washed in 200 µl of RNase free 70% ethanol before being centrifuged at maximum speed for 5 min at 4 °C. Supernatant was removed and the pellet air-dried slightly before being resuspended in RNase-free water.

## Live dye and drug treatments

Fluorescent styryl molecules, RNA-selective probes F22 and E36 (kindly provided by Young-Tae Chang[20]) were microinjected at 0.25 µM. The ribosomal-RNA selective live probe, rRNA-naphthalimide, was synthesized as detailed in ref. 29 (protocol kindly provided by Dr. Peng Wei) and microinjected at 0.25 µM.

For experiments using LysoSensor™ Green DND-189 (ThermoFisher, L7535) embryos were cultured in 1 nm LysoSensor™ Green DND-189 diluted in KSOM media and incubated for 15 min at 37 °C and 5% $CO_2$. Embryos were washed 3 times in pre-warmed KSOM before being transferred to imaging chamber for live imaging.

For drug treatments, all drugs were diluted in KSOM. Nocodazole (Sigma, M1404) was used at 70 µM for 3 h[48], Ciliobrevin D (Merck, 250401) was used at 70 µM for 3 h[48,69]. Puromycin (Sigma, P8833) was used at 50 µg for 2 h and Cycloheximide (Sigma, C7698) was used at 20 µg for 5 min and 2 h, as described[70,71] to block translation.

## Immunofluorescence

Embryos were fixed with 4% paraformaldehyde (PFA) in 1x phosphate-buffered saline (PBS) or 1.5% PFA in Hanks' Balanced Salt solution (HBSS) for CAMSAP3 staining for 30 min, then rinsed once in 1x PBS before permeabilization using 0.5% Triton X-100 in 1x PBS for 30 min to 1 h. Embryos were blocked with 10% FBS in 1x PBS (10% FBS-PBS) for 1 h and primary antibodies incubated for ~ 20 h at 4 °C diluted in 10% FBS-PBS at the following concentrations: β-tubulin at 1:1000 (Abcam, ab21057), CAMSAP3 at 1:500 (Aviva OAAB13856), CAMSAP3[72] at 1:500, Ninein at 1:500 (MilliPore, MABT29), LAMP1 at 1:1000 (Abcam, ab25245), α-tubulin at 1:1000 (Sigma, T6199), acetylated α-tubulin at 1:1000 (Sigma, 6793), Sec61β at 1:500 (Cell Signalling, 14648), CDX2 at 1:100 (Abcam, ab88129), CDX2 at 1:100 (Abcam, ab157524), Oct3/4 at 1:100 (Santa Cruz, sc5279), SOX2 at 1:200 (ThermoFisher, 14-9811-82). After primary antibody incubation, embryos were washed three times with 1x PBS for 10 min before adding secondary antibodies for 2 h diluted to 1:500 in 10% FBS-PBS. Secondary antibodies used in this study were goat anti-rat IgG AF647 (Invitrogen, A21247), donkey anti-

mouse IgG AF488 (Invitrogen, A21202), donkey anti-rabbit IgG AF586 Invitrogen, A10042), donkey anti-rabbit IgG AF488 (Invitrogen, A21206), goat anti-rat IgG AF488 (Invitrogen, A11006), donkey anti-goat IgG AF647 (Invitrogen, A32849) at 1:500. 4',6-diamidino-2-phenylindole (DAPI; Sigma, D1306) was used at a concentration of 5 µg/ml, Phalloidin-Rhodamine (Molecular Probes, R415) at 1:500 and RNA-selective dye F22 at 25 mM and were added with secondary antibodies. Embryos were washed again three times in 1x PBS for 10 min each. For imaging, embryos were transferred into a 10 µl 1x PBS drop overlayed with 300 µl mineral oil in LabTek chambers (ThermoFisher, NUN155411). Immunofluorescence imaging was performed using a Zeiss LSM780 confocal or Zeiss LSM980 Airyscan microscope.

## RNase digestion
Embryos were first treated with RNA dye F22 at 0.25 µM prior to fixation in 4% PFA, permeabilised and blocked (as detailed above). Embryos were also labelled for microtubules using acetylated α-tubulin at 1:1000 (Sigma, 6793) and nuclei with DAPI (Sigma). Embryos were then imaged prior to RNase treatment. Subsequently, embryos were incubated in 25 µg/ml of RNase (Ribonuclease A, Sigma, R6513) in 1x PBS at 37 °C in 5% $CO_2$ for 3 h. Control embryos were incubated in 100 µl of 1x PBS without RNase. Embryos were rinsed in 1x PBS before being imaged a second time, using the same settings of the first imaging. Acetylated α-tubulin and DAPI staining was used as an internal control to match embryos before and after RNase treatment.

## Poly(A) RNA fluorescent in situ hybridisation (FISH)
Poly(A) RNA FISH was adapted from published smFISH protocols[73,74]. Embryos microinjected with the RNA-F22 probe were fixed for 30 min in 4% PFA, then permeabilised in 0.5% Triton X-100 for 30 min. Embryos were then washed with 2x saline sodium citrate buffer (SSC) (Sigma, 36639-1L) containing 10% (v/v) formamide. Before hybridisation, embryos were transferred to hybridisation buffer (2x SSC, 10% (v/v) formamide, 100 mg/mL dextran sulfate (Sigma, 42867-5 G), 1 mg/mL yeast RNA), and pre-incubated for 1 h at 37 °C. Fluorescently labelled DNA oligonucleotide probe, dT(18)-Fluorescein HPLC-purified (Integrated DNA Technologies (IDT)) was diluted into hybridisation buffer and applied to embryos at a final concentration of 400 nM. After 16 h at 37 °C in a slightly shaking incubator, embryos were washed with 2x SSC, 10% formamide, incubated again at 37 °C for 1 h, before washing into 2x SSC. Embryos stained with DAPI diluted in 2x SSC and incubated for 30 min. After incubation embryos were washed in 2x SSC and transfer to chamber for imaging.

## RNA fluorescent in situ hybridisation (RNA FISH)
RNA FISH was adapted from ref. [68]. Embryos microinjected with the RNA-F22 probe were fixed for 30 min in 4% PFA then permeabilised in 0.5% Triton X-100 for 30 min. Embryos were dehydrated through graded methanol/PBS series with 2 min washes at room temperature; 25% Methanol/PBS, 50% Methanol/PBS, 75% Methanol/PBS, 2 × 100% Methanol. Embryos were then rehydrated through a graded methanol/PBS series with 2 min washes at room temperature; 75% Methanol/PBS, 50% Methanol/PBS, 25% Methanol/PBS. Embryos were then washed three times for 3 min in 0.3% Triton X-100/PBS. Embryos were transferred to prehybridisation solution (50% Formamide (Sigma, F7503), 5x SSC, 2% blocking powder (Roche, 11096176001), 0.1% TritonX-100, 1 mg/mL yeast RNA (Invitrogen, AM7119), 5 mM EDTA, and 50 mg/mL heparin (Sigma, H3149), made up in RNase-free water. Stored at −20 °C and preheated to 65 °C immediately prior to use) and incubated for 2 h at 65 °C in a slightly rocking incubator. Fluorescein-labelled riboprobes were diluted in prehybridisation solution at 0.5 ng/µl and incubated overnight at 65 °C. Embryos were then washed through a graded Solution 1/2x SSC series for 3-min washes at 65 °C; 100% Solution 1 (50% Formamide, 5x SSC, 0.5% CHAPS and 0.1% TritonX-100 made up in RNase-free water. Stored at room temperature and preheated to 65 °C immediately prior to use); 75% Solution 1/25% 2x SSC; 50% Solution 1/50% 2x SSC; 25% Solution 1/75% 2x SSC. Embryos were then washed with 2x SCC at 65 °C for 15 min each wash and subsequently transferred to 2x SSC for imaging.

## tRNA fluorescent in situ hybridisation (tRNA FISH)
tRNA FISH (tFISH) was adapted from published tRNA FISH protocols[27,28,75]. Embryos microinjected with the RNA-F22 probe were fixed for 30 min in 4% PFA, then permeabilised in 0.5% Triton X-100 for 30 min. Embryos were then dehydrated through a graded methanol/PBS series; 70, 90, and 100% methanol successively for a duration of 5 min each. Embryos were then incubated in pre-hybridisation buffer (containing 10% dextran sulfate, 2x SSC, 125 µg Escherichia coli tRNA/ml) for 2 h at 37 °C. Fluorescently-labelled tRNA probe-fluorescein HPLC-purified (Integrated DNA Technologies (IDT)) was diluted into prehybridisation buffer and applied to embryos at a final concentration of 0.5 ng/ml. Hybridisation was carried out at 37 °C overnight in a slightly rocking incubator. Embryos were then washed three times with 2x SSC at 45 °C for tRNA. Finally, embryos were then washed three times for 10 min with 1x SSC at room temperature before being transferred to an imaging chamber containing 1x SSC for imaging.

## Ribopuromycylation method (RPM)
A ribopuromycylation method (RPM) was performed to examine actively translating ribosomes using a protocol adapted from refs. [33–35]. Embryos were incubated in pre-warmed KSOM media containing Puromycin (Sigma-Aldrich, P7255) at 50 µg/ml, Emetine at 25 µg/ml for 5 min at 37 °C, 5% $CO_2$. Control embryos were incubated in puromycin without Emetine. Embryos were subsequently washed in KSOM before being transferred to co-extraction/fixation buffer (0.015% (m/v) Digitonin (Sigma, D141-100MG), 50 mM Tris-HCl pH 7.5, 5 mM $MgCl_2$, 25 mM KCl, 0.2 M sucrose, 355 µM cycloheximide, 1x EDTA-free protease inhibitors (1 tablet per/10 ml), 10 U/ml RNase Out, 3% PFA, DEPC-treated water (Ambion, AM9902) adjusted to pH 7.4) and incubated for 20 min at room temperature.

Following co-extraction/fixation, embryos were immediately fixed using 4% PFA for 10 min at room temperature and then rinsed once in PBS before proceeding with immunostaining. Embryos were incubated in staining buffer (0.05% Saponin, 10 mM Glycine (MP Biomedicals, 100570), 5% FBS-PBS) for 15 min at room temperature. Subsequently, primary antibodies diluted in staining buffer at the following concentrations: RPL10A at 1:1000 (ABclonal, A20944) and anti-puromycin clone 12D10 at 1:300 (Merck, MABE343) were incubated for 16–18 h at 4 °C.

After primary antibody incubation, embryos were washed in staining buffer before adding secondary antibodies for 2 h diluted to 1:500 in staining buffer. Embryos were washed again three times in 1x PBS for 10 min each before being subject to imaging.

## Confocal microscopy
Live embryo imaging was performed as described previously[9,48]. Embryos were transferred into a 10 µl 1x KSOM drop overlayed with 300 µl mineral oil in LabTek chambers (ThermoFisher, NUN155411). For confocal imaging a laser scanning confocal (LSM780, Zeiss) with water UV-VIS-IR Apochromat 63× 1.2 NA objective, Immersion Oil W 2010 (Zeiss, 444969-0000-000) and avalanche photodiode light detectors (APDs) of the Confocor 3 module was used, fitted with an incubation chamber for controlled conditions at 37 °C and 5% $CO_2$. Images of entire 3D embryos were acquired as a reference by setting the z-stack using the following parameters: image size 512 × 512, pinhole 3–3.2 µm, pixel dwell time 6.30 µm/s, at 1.5 µm section intervals. For two channel imaging APDs were used with the 488 and 561 laser lines combined in the same track with appropriate filter settings depending on the fluorophores. For multi-channel imaging additional laser lines 405 and

647 were added to the same track using GaAsP detectors and photo-multiplier tubes (PMTs).

For long-term time-lapse experiments, images were acquired using the following parameters: 512 × 512, pinhole 3.2 μm, laser power range 0.2–2.5%, pixel dwell time 6.3 μm/s and z-stack was acquired with 1.5 μm section intervals. Images were acquired at intervals ranging between 5 and 30 min depending on experimental design.

For high spatiotemporal resolution time-lapse images, regions of interest were identified from 3D z-stack images of entire embryos. Zoom was increased up to 4× and using the region of interest module tool in Zen Black software a rectangular box was drawn around the selected region of interest. Using a pinhole range between 2-3 μm, pixel resolution 512 × 512 or 1024 × 1024 and pixel dwell time of 6.30 μm/s, images were acquired at 500 ms-2 s intervals in a single 1 μm z-plane using the time-series function.

For imaging using the Zeiss LSM980 (relating to Fig. 2d, k; Fig. 3c, d, f, g; Fig. 4a, c, g, h; Fig. 5a–d; Fig. 6a, c, f, h; Fig. 7d, h; Fig. 9a, c and Supplementary Fig. 3d, and Supplementary Fig. 4c, f, k, l–p) embryos were imaged using Airyscan 2 detector, 32 + 2 spectral GaAsP detector with two flanking PMT's and transmitted light detector photo-multiplier tubes (T-PMPT). The Axio observed. Z1/7 and C Plan-Apochromat 63x/1.40 oil objective with Optovar Tubelens LSM was used. Images of entire 3D embryos were acquired by setting the z-stack using the following parameters: image size 2248 × 2248, 0.043 μm × 0.043 μm × 0.170 μm scaling per pixel, z-stack slices ~420 for total embryo diameter, pinhole 5.12 arb. units/312 μm, zoom 1.4x, pixel dwell time 1.86 ms with frame times up to ~6 s. For two channel imaging GaAsp-PMT detectors were used in Airyscan mode (Zeiss). For imaging of three channels images were acquired at 2372 × 2372 pixels, 312 μm pinhole, zoom 1.6x, 0.33 ms pixel time and 17.58 s with frame time at 17.58 s and, LSM scan speed at 8, 8x averaging. Image processing was completed using the Airyscan processing module in ZEN Blue 3.3 software (Zeiss).

For imaging using the Leica Stellaris5 Invert Confocal Microscope (relating to Fig. 4e and Supplementary Fig. 4f), embryos were imaged using 63x/1.2 oil WD 140 μm immersion objective. rRNA-naphthalimide was excited using the 488-nm laser, and RNA-F22 was excited using the 561-nm laser. Images of entire 3D embryos were acquired using LAS X (Leica) and by setting the z-stack using the following parameters: image size 512 × 512, pinhole 3 μm, pixel dwell time 6.30 μm/s, at 1.5 μm section intervals. For quantitative analyses, laser power and detector gain were maintained constant.

## Lattice light sheet microscopy

Embryos were transferred into a 10 μl 1x KSOM drop overlaid with 400 μl mineral oil in ibidi m-slide 8-well glass bottom chambers (IBIDI GMBH, 80827). For lattice light-sheet imaging (relating to Fig. 8a), we employed the Zeiss Lattice Light Sheet 7 (LLS7 - Zeiss – Pre-serial) ZEN blue 3.4 software (Zeiss). Time-lapse imaging was acquired using light sheets (488 nm and 633 nm) of 100 μm length with a thickness of 1.8 μm were created at the sample plane via a 13.3× 0.44 NA objective. Fluorescence emission was collected via a 44.83x, 1 NA detection objective. Aberration correction was set to a value of 182 to minimise aberrations as determined by imaging the Point Spread Function using 100 nm fluorescent microspheres at the coverslip of a glass bottom chamber slide.

Data were collected with a range of frame rates of 1 ms to 20 ms and a y-step interval of 600 nm. Light was collected via a multi-band stop, LBF 405/488/561/633, filter. Data was subsequently deskewed then deconvolved using a constrained iterative algorithm and 20 iterations in ZEN blue 3.5 software (Zeiss).

## Serial blockface scanning electron microscopy (SBF-SEM)

Embryos were fixed in 2.5% glutaraldehyde in embryo grade water (Sigma, W1503) for 1–2 h at room temperature. Fixed embryos were rinsed with 1x PBS and stored at 4 °C before processing. Processing was performed exactly as described previously[76]. Serial blockface sectioning and scanning electron microscopy were performed using a VolumeScope serial block-face EM (Thermo Fisher, Waltham, MA) equipped with a low-vacuum backscatter detector (VS-DBS; Thermo Fisher).

## ChIP-seq preprocessing and peak calling

We obtained single-end (SE) fastq files from Adachi et al., with the following accession numbers: GSM703186 (Sox2 ESC) and GSM1246724 (Cdx2 TSC)[40]. The raw fastq files were quality trimmed with trimmomatic v0.36[77] using parameters SE -phred33 SLI-DINGWINDOW:3:15 MINLEN:36. The quality trimmed fastq files were then aligned with Bowtie2 v2.3.4.1[78] using the SE parameter -U. Sequence alignment map (.sam) files were compressed into binary files (.bam) using sambamba v0.6.7[79] using default parameters. Sambamba was then utilised, with default parameters to sort and index the .bam files. The .bam files were then filtered for blacklisted regions using samtools view v1.12[80], duplicates were removed using Picard v2.18.0 with parameters REMOVE_DUPLICATES=true, ASSUME_SORTED=true, VALIDATION_STRINGENCY = LENIENT and CREATE_INDEX=true. Low quality reads were removed using samtools view v1.12[80], with parameter -q 10. The base function GNU parallel v20161222[81] was used for parallel processing of samtools view. deepTools bamCoverage v3.4.3[82] was used to generate genome coverage tracks (bigwig format) from the processed .bam files using the parameters -bs 30 –normalizeUsing RPKM –effectiveGenomeSize 2652783500 -of bigwig. UCSC genome browser[83] was used for visualisation of the bigWig tracks.

## Analysis of ChIP-seq targets

Pre-calculated z-scores for expression throughout preimplantation embryo development were utilised from Boroviak et al., scRNA-seq data[84]. A custom script with R (v4.2.1) base functions[85] and ggplot2 v3.4.0[86] were used for data exploration and visualisation. Expression correlation values were calculated with R base function cor using default parameters, and were visualised with pheatmap v1.0.12[87] using default parameters.

## Image and statistical analysis

Image analyses were performed using Imaris 7.4.2 and 9.5.1 software (Bitplane AG), Zen Blue and Black software (Zeiss) and ImageJ/Fiji software. For 3D segmentation of embryos and cells, the Imaris manual surface rendering module and mask function was used. Fluorescence intensities in apical and basal regions were measured by segmentation of full 3D cells, the sum fluorescence intensities were obtained using the Imaris statistic function. Fluorescence intensity plot profiles were generated on 2D surfaces using the Fiji multi-plot function, the mean fluorescence intensity values and corresponding X and Y coordinates were utilised for analysis. Fluorescence intensities were normalised to cell volume and Membrane-GFP, BFP-Utrophin or DAPI where applicable.

Pixel percentage co-occurrence analyses were performed using Fiji/ImageJ. In brief, image threshold was determined to reduce background signal and kept consistent across all images analysed. Binary images were created and converted to mask, particle analysis was then performed to measure the area signal of pixel overlay in channels of interest. Image math was performed using the Image Calculator functions 'AND' and 'XOR' to determine the percentage area of pixel co-occurrence.

RNA foci dynamics were quantified and analysed using the Imaris manual spot-tracking module. Only foci which were detected in at least 4 continuous time frames were used for analysis. RNA distance was quantified using the Imaris manual measurement module.

Any adjustments made to images were applied to entire images and remained consistent across each experiment. To enhance the accessibility of scientific figures, individual channels were pseudo-

coloured. Whenever possible, a colour combination of magenta and cyan/teal, or white and magenta was utilised.

### Statistics and reproducibility

Reproducibility was confirmed by independent experiments. Statistical analyses were performed using GraphPad Prism software (http://www.graphpad.com) and Excel (Microsoft Office). If data showed a normal distribution, the comparison of two samples were completed using unpaired two-tailed Student's t-test or one-way ANOVA test for multiple groups. For data without normal distribution Mann-Whitney two-tailed t-test two experimental groups or Kruska–Wallis test was used. Statistical significance was defined using p value style: 0.1234 (ns), 0.0332 (*), 0.0021 (**), < 0.0001(****). Box plots display minimum, lower quartile, median, upper quartile and maximum. Column and line graphs show mean + standard error of the mean (SEM). $n$ represents number of cells otherwise specified in figure legends. Adobe softwares (Illustrator and Photoshop 2023) were used to generate figures and schematics.

### Reporting summary

Further information on research design is available in the Nature Portfolio Reporting Summary linked to this article.

## Data availability

The authors declare that the data supporting the findings of this study are available within the article and Supplementary Information files. Supplementary movies are linked to this article. Source data are provided with this paper as a Source data file. All remaining data will be available from the corresponding author upon reasonable request. Source data are provided with this paper.

## Code availability

Codes may be accessed at the github repository: https://github.com/pololab/Hawdon_etal_2023. Datasets can be accessed using GSM703186 and GSM1246724.

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

## Acknowledgements

We acknowledge the scientific and technical support of Monash Micro Imaging Facility (MMI) and Monash Animal Research Platform (MARP) at Monash University as well as the use of the Microscopy Australia Research Facility at the Center for Microscopy and Microanalysis at the University of Queensland. We thank Prof. Young-Tae Chang for providing the RNA-selective live cell imaging probes and Dr. Peng Wei for providing the rRNA synthesis protocol. We thank A/Prof. Jan Kaslin, Alon Douek, Prof. Christina Mitchell, Prof. Masatoshi Takeichi, Prof. Elizabeth New, and A/Prof. Robin Hobbs for providing reagents, and Dr. Marcel Sayre for assistance with SBF-SEM. This work was supported by the National Health and Medical Research Council (NHMRC) Ideas Grant APP2002507 and Investigator Grant APP2009409 to J.Z. J.Z. was also supported by the Sylvia & Charles Viertel Senior Medical Fellowship and the Canadian Institute for Advanced Research (CIFAR) Azrieli Scholarship. J.M.P. was supported by the NHMRC Ideas Grant APP2004627. R.G.P. was supported by the NHMRC Ideas Grants APP1140064 and APP1150083, NHMRC fellowship APP1156489 and an Australian Research Council (ARC) Laureate Fellowship. The Australian Regenerative Medicine Institute is supported by grants from the State Government of Victoria and the Australian Government. The South Australian immunoGENomics Cancer Institute (SAiGENCI) received grant funding from the Australian Government.

## Author contributions

A.H. and J.Z. conceived and designed the study. A.H. and J.Z. performed experiments, except for Lattice Light Sheet imaging was performed by N.G., ChIP-seq data analysis was performed by M.M. and J.M.P., and serial blockface scanning electron microscopy was performed by C.F. and R.P. A.H. and A.E. performed data analyses. A.H. and J.Z. wrote the manuscript. All authors edited and approved the manuscript.

## Competing interests

The authors declare no competing interests.
