## [Peer Review File · Nature Communications]

Apicobasal RNA asymmetries regulate cell fate in the early mouse embryoREVIEWER COMMENTS

Reviewer #1 (Remarks to the Author):

In this manuscript, Hawdon and colleagues use live cell microscopy to monitor bulk RNA localization within developing mouse embryos. They find that in late 16 cell-stage embryos, the bulk of cellular RNA is located toward the basal end of outer cells, leading to asymmetric inheritance of RNAs in cells that divide parallel to the basal membrane. Conversely, they find that translation is more active toward the apical end of the cell with translation initiation factors and a ribosomal protein apically enriched. They define a role for lysosome-assisted RNA transport in setting up the basal RNA enrichment and find that the lysosome-assisted transport may be important for embryo development.

Overall, the manuscript relates interesting findings and uses sophisticated microscopy methods to explore an exciting area of research. I do, however, have technical concerns and question if some conclusions are directly supported by the data presented. These are outlined below.

MAJOR COMMENTS

1. My biggest concern with the entire manuscript lies with the use of styryl probe F22 as an RNA marker and the conclusions that are drawn. In most cells, ~90% of the cellular RNA is rRNA with mRNA comprising 2-5% of the total. F22, unsurprisingly, readily binds rRNA. In fact, in the development of F22, nucleolar RNA signal (presumably representing rRNA) was the major source of fluorescence (PMID 16793519 figure 4). If we assume that the vast majority of the F22 signal throughout this manuscript is rRNA (as the authors themselves say might be the case in the discussion), this makes it difficult to reconcile specific results as rRNA in the cytoplasm is essentially always associated with ribosomes.

For example, if the vast majority of F22 signal (and therefore rRNA...and therefore ribosomes) is at the basal end of these cells, how is the apical end more translationally active? How is RpL10 similarly apically enriched? It seems to me at least that the idea that almost all of the F22 signal potentially reflecting ribosomes should be more thoroughly discussed and dissected as it changes how results are interpreted.

To be clear, I don't necessarily mean that the binding of F22 to rRNA invalidates results, but that it should be more thoroughly considered when interpreting them.

2. In figure 2, the authors propose that RNA is transported basally along microtubules to their plus ends in these cells. Part of the evidence they use for this comes from the finding that inhibiting dynein-facilitated transport enhanced basal RNA accumulation. This is fine, but it seems odd to not more directly test this by inhibiting kinesins.

3. In figure 4, the authors argue that the apical compartment is more translationally active. One piece of evidence they present is that upon inhibiting translation with cycloheximide, RNAs in the apical compartment show an increase in speed, which they argue is due to ribosomes leaving the RNA, allowing it to move more freely. However, unlike puromycin, cycloheximide stabilizes ribosomes on mRNA, freezing them in place. It's therefore unclear how this would be expected to make RNA apparently more mobile.

4. In figure 5, the authors argue that the observed RNA asymmetry contributes to embryo development as disturbing the asymmetry using ANXA11 knockdown perturbs development. This may be true, but in addition to disturbing RNA distribution, ANXA11 knockdown also disturbed lysosome distribution (extended data fig 3D), so it's difficult to directly ascribe observed effects only to perturbing asymmetric RNA inheritance.

MINOR COMMENTS

1. In line 149, the authors say "Directed transport of RNA cannot occur by diffusion but requires the guidance of the microtubule cytoskeleton..." This is not quite true. For example, Nos RNA in the *Drosophila* embryo is localized through cytoplasmic streaming / diffusion and then anchoring at the site of localization. In yeast Ash1 RNA is localized to daughter cells through an actin-dependent, not microtubule-dependent, system. There even need not be directed transport at all to establish RNA gradients. In *Drosophila* embryos, Hsp83 RNA gradients are established through spatially defined regions of RNA protection and degradation.

2. There are places where the wording overstates conclusions that can be drawn. For example, in line 165, the authors say that "As the embryos progresses from early to mid 16-cell stage the microtubule network expands, and at the same time the RNA moves basally, indicating a highly spatiotemporally regulated microtubule-dependent transport of RNA at the time of the first embryonic cell fate decision." These two findings alone do not indicate that. They simply indicate that these two things are happening at the same and therefore may be connected. To directly indicate microtubule-dependent transport, the system should be specifically perturbed, which the authors do later with Ciliobrevin, but not here.

Reviewer #2 (Remarks to the Author):

The paper by Zenker and colleagues examines the distributions of apicobasal RNA and its role in cell fate in the early mouse embryo. The study relies heavily on the usage of the RNA-selective fluorescent dyes F22 and E36 that were synthesized for visualizing RNA. The caveat for RNA-selective dyes in general is that it is unclear what type of RNA is being looked at because the dye may label rRNA, tRNA, or even mitochondria RNAs (their mRNAs, rRNA, tRNA, etc.). As the focus of this study presumably is on nuclear-encoded cytosolic mRNA, the authors need to employ another method to verify that the fluorescent puncta they are studying are actually mRNA. For example, they compared the ER density with the presence of Cdx2 and SOX2, so smFISH of mRNAs controlled by these factors could be done.

The evidence provided for increased translation is inadequate. Their evidence of translation is the accumulation of factors required for translation (i.e. ribosomal protein L10A, PolyA binding protein, eIF2beta subunit, and ER) rather than the actual translation itself. To my eyes, these factors seem to surround the "granule-looking" RNA signals rather than being co-localized within the granule. This would suggest the granules are devoid of translation factors. Even if co-localized in stress granules, some ribosomal subunits and other initiation factors and PABPs may associate with the granule but not necessarily engage in translation. To demonstrate that translation is occurring in granules, assays showing translation using anti-puromycylation after puromycin are recommended.

The role of annexin A11 for RNA trafficking to the basal location of the outer blastomere, whose motion appears to be dependent on microtubules, is exciting. To demonstrate the RNA signal in these experiments represent RNA granules, the authors need to stain for G3BP1 and/or DBP1.

With additional work to address the points above, the paper can be made suitable for publication in Nature Communications.

Reviewer #1 (Remarks to the Author):

In this manuscript, Hawdon and colleagues use live cell microscopy to monitor bulk RNA localization within developing mouse embryos. They find that in late 16 cell-stage embryos, the bulk of cellular RNA is located toward the basal end of outer cells, leading to asymmetric inheritance of RNAs in cells that divide parallel to the basal membrane. Conversely, they find that translation is more active toward the apical end of the cell with translation initiation factors and a ribosomal protein apically enriched. They define a role for lysosome-assisted RNA transport in setting up the basal RNA enrichment and find that the lysosome-assisted transport may be important for embryo development.

Overall, the manuscript relates interesting findings and uses sophisticated microscopy methods to explore an exciting area of research. I do, however, have technical concerns and question if some conclusions are directly supported by the data presented. These are outlined below.

MAJOR COMMENTS

1. My biggest concern with the entire manuscript lies with the use of styryl probe F22 as an RNA marker and the conclusions that are drawn. In most cells, ~90% of the cellular RNA is rRNA with mRNA comprising 2-5% of the total. F22, unsurprisingly, readily binds rRNA. In fact, in the development of F22, nucleolar RNA signal (presumably representing rRNA) was the major source of fluorescence (PMID 16793519 figure 4). If we assume that the vast majority of the F22 signal throughout this manuscript is rRNA (as the authors themselves say might be the case in the discussion), this makes it difficult to reconcile specific results as rRNA in the cytoplasm is essentially always associated with ribosomes.

We thank reviewer 1 for this comment, and for finding our work significant and interesting. We agree that the RNA styryl probes (F22 and E36) are described to preferably bind to rRNA (Li et al., 2006). To determine if F22 labels exclusively rRNA or also to some extent other RNA subtypes, including mRNA and tRNA (most relevant RNA subtypes for translation events, the topic of our manuscript) in the early mammalian embryo, we performed co-labelling experiments of F22 with:

- 1) Poly(A) RNA FISH using a polydT(18)-Fluorescein probe to label specifically mRNA (new Fig. 4a, b, Supplementary Fig. 4a)
- 2) a fluorescein-tagged oligonucleotide complementary to mammalian tRNA^{Lys} to label specifically tRNA (Shaheen and Hopper, 2005; Shaheen et al., 2007) (new Fig. 4c, d, Supplementary Fig. 4b, c), called tFISH
- 3) rRNA-naphthalimide to label specifically rRNA (Cao et al., 2019) (new Fig. 4e, f, Supplementary Fig. 4d)

While the majority of F22 signal overlays with rRNA-naphthalimide (new Fig. 4e, f, Supplementary Fig. 4d), there are additional F22 foci which are negative for rRNA-naphthalimide, particularly in apical regions of 16-cell stage outer blastomeres. This result demonstrates that F22 signals may contain other RNA subtypes than rRNA. Using poly(A) RNA FISH and tFISH, we identified F22 signals in apical areas to be co-labelled for mRNA and tRNA (new Fig. 4a-d, Supplementary Fig. 4a-d). Thus, these experiments support our findings that 1) F22 labels global RNA transcripts, including rRNA but also mRNA and tRNA, 2) that there is a higher presence of all 3 RNA

subtypes (rRNA, mRNA and tRNA) in apical regions compared to basal regions, where 3) higher contents of rRNA transcripts can be found.

Having explored that F22-labelled RNA foci contain global mRNAs, we extended our studies to visualise target messenger RNA transcripts in F22-labeled RNA foci. We chose eIF2 β as an obvious candidate due to its essential role in translation initiation. In line with its apical protein localisation, eIF2 β mRNA is located in apical RNA foci (new Supplementary Fig. 4l-n).

For example, if the vast majority of F22 signal (and therefore rRNA...and therefore ribosomes) is at the basal end of these cells, how is the apical end more translationally active? How is Rpl10 similarly apically enriched?

We thank reviewer 1 to allow us to further explain the opposing accumulation of rRNA in basal regions and of ribosomal protein RPL10A, coinciding with translation capacity, in apical regions.

First, we would like to answer “*how is Rpl10 similarly apically enriched?*”

The coordination of rRNA and ribosomal protein synthesis is differentially regulated in stem cells as compared to differentiated cells. Naïve pluripotent cells have a higher rRNA content but lower ribosomal protein content, especially for large ribosomal subunit components, including RPL10A (Hayashi et al., 2014; Sanchez et al., 2016; Watanabe-Susaki et al., 2014; Zhang et al., 2014). Following rRNA reduction, stem cells exit pluripotency and differentiate (Woolnough et al., 2016). This is regulated by the RNA-binding protein HIV-1 Tat Specific Factor 1 (HTATSF1), which is upregulated during naïve pluripotency (Corsini et al., 2018). HTATSF1 causes the retention of introns specifically in ribosomal proteins and thus, a lower ribosomal protein synthesis. Contrary, HTATSF1 triggers rRNA maturation by removing the ITS1 sequence from rRNA (Corsini et al., 2018). In line with this, we observe a higher expression of HTATSF1 in basal regions as compared to apical regions in outer blastomeres of 16-cell stage embryos (new Fig. 4h and Supplementary Fig. 4e) explaining a high rRNA-low RPL10A content in basal regions but a low rRNA-high RPL10A content in apical regions. Therefore, in line with recent studies on stem cell cultures (Corsini et al., 2018; Woolnough et al., 2016), we show here a similar prerequisite of high rRNA content for pluripotency in the cells of the living embryo.

Next, we would like to explain “*how is the apical end more translationally active?*”

While ribosomes are the sites of translation, a complex multi-step mechanism must occur for translation to take place.

Briefly, the initiator tRNA, delivered by eukaryotic initiation factor eIF2 (Beilstein-Edmands et al., 2015) of which subunit eIF2 β is accumulated at apical RNA foci in 16-cell stage outer blastomeres of the early mouse embryo (Fig. 4c, now Fig. 4k), binds the small ribosomal subunit at the 5' cap of mRNAs. As shown with our new experiments outlined above (new Fig. 4a - f), the vast majority of mRNA and tRNA subtypes are also located apically. Only after this tripartite complex (initiator tRNA,

eIF2 and small ribosomal subunit) is formed, the large ribosomal subunit joins to form a complete ribosome. We have now added a co-labelling experiment for the large ribosomal subunit (L10A-eGFP) and the small ribosomal subunit (RSP6-mScarlet), which shows a higher presence and rate of co-localisation of both subunits in apical regions compared to basal regions in outer blastomeres of late 16-cell stage embryos (new Fig. 4g and Supplementary Fig 4e). Thus, while the amount of rRNA is higher at basal regions, the number of joint ribosomal subunits, equipped with essential co-factors for translation to occur, is higher in apical regions.

To demonstrate active translation in subcellular areas of 16-cell stage blastomeres, we 1) performed a ribopuromycylation method (RPM) as kindly suggested by reviewer 2 (please see below for more details) (new Fig. 4n-q and Supplementary Fig. 4f-h) and 2) we adopted the SunTag assay on late 16-cell stage embryos (Tanenbaum et al., 2014; Yan et al., 2016) (new Fig. 4s-u and Supplementary Fig. 4i-k and o,p).

The RPM assay allowed us to detect global translation events independent of the gene-specificity of mRNA transcripts on fixed samples. It demonstrates a higher rate of ribosome-attached puromycylated nascent peptides in apical as compared to basal regions (new Fig. 4n, o) in outer 16-cell stage blastomeres.

The SunTag system, however, differs in being applied on live samples to visualise the translation rates of mRNAs encoding a specific gene. We choose to analyse the real-time translation dynamics of translation initiation factor eIF2 β , an important factor in our manuscript to demonstrate translation capacity (new Fig. 4s-u and Supplementary Fig. 4i-p). The mRNA and nascent protein of eIF2 β was predominantly located apically and in some instance the two signals overlaid showing actual translation until splitting apart.

In conclusion, the asymmetric recruitment of translation co-components, including mRNA, tRNA, ER, PABPC1 and eIF2 β , with spatiotemporal precision and the HTATSF1-dependent regulation of rRNA and ribosomal protein content enables a higher translation rate in apical regions as shown by two independent technologies, the RPM and SunTag.

It seems to me at least that the idea that almost all of the F22 signal potentially reflecting ribosomes should be more thoroughly discussed and dissected as it changes how results are interpreted. To be clear, I don't necessarily mean that the binding of F22 to rRNA invalidates results, but that it should be more thoroughly considered when interpreting them.

We understand the reviewer's concern and it is very important to us that our results are very clearly presented. Therefore, we decided that the best way to do so is with a summarising schematic presentation which can now be found as new Fig. 5. It shows where the various factors and components are located at late 16-, 16- to 32- and 32-cell stage. We are happy to add more details if desired.

2. In figure 2, the authors propose that RNA is transported basally along microtubules to their plus ends in these cells. Part of the evidence they use for this comes from the finding that inhibiting dynein-facilitated transport enhanced basal RNA accumulation. This is fine, but it seems odd to not more directly test this by inhibiting kinesins.

We thank reviewer 1 for this suggestion. The experiment proposed to inhibit kinesin-facilitated transport is interesting, however, we respectfully would like to outline the technical limitations to do so.

The experiment requires a cell-permeable small molecule compound to enable inhibition with temporal precision at 16-cell stage. Other tools, for instance a pan-kinesin blocking antibody or siRNAs against individual kinesin motors, to be microinjected into the living mouse embryo at 1-cell stage will cause developmental arrest of the embryo even before reaching 16-cell stage due to the ubiquitous functionality of kinesin motors, particularly during cell division.

There is one cytoplasmic dynein encoded in the mammalian genome (Dynein 1), besides 8 axonemal dyneins which have no functionality during preimplantation embryogenesis due to the absence of cilia and flagella (Roberts et al., 2013). Thus, our approach using Ciliobrevin D allowed us to specifically target the only microtubule minus end-directed motor in the cells of the preimplantation embryo at the exact time to specifically interfere with RNA transport.

In contrast, there are 45 kinesin family members expressed in the mouse (Kalantari and Filges, 2020). Therefore, we believe that the characterisation of the kinesin motor(s) facilitating basally directed transport of RNA at 16-cell stage during mouse preimplantation embryogenesis is beyond the scope of the current manuscript. Furthermore, to the best of our knowledge, there are small compounds available against only 4 kinesin motors to date with suboptimal properties to perform the proposed experiment in living mouse embryos:

- 1) Kinesore (Randall et al., 2017), an activator and not an inhibitor of kinesin 1-dependent microtubule transport
- 2) Paprotrain (Labrière et al., 2016), which blocks specifically Mitotic kinesin-like protein 2 (MKLP-2), thereby inhibiting cellular Aurora B and Survivin anaphase centromere-to-spindle midzone relocation. Thus, MKLP-2 is very unlikely the major kinesin involved in RNA transport during 16-cell stage interphase.
- 3) Monastrol (Kapoor et al., 2000), inhibitor of mitotic kinesin Eg5 which cause mitotic arrest due to spindle mis-assembly. Thus, Eg5 is very unlikely the major kinesin involved in RNA transport during 16-cell stage interphase.
- 4) BTB-1 (Catarinella et al., 2009), inhibitor of mitotic motor Kif18A, a key regulator of chromosome aggregation and stability. Thus, Kif18A is very unlikely the major kinesin involved in RNA transport during 16-cell stage interphase.

3. In figure 4, the authors argue that the apical compartment is more translationally active. One piece of evidence they present is that upon inhibiting translation with cycloheximide, RNAs in the apical compartment show an increase in speed, which they argue is due to ribosomes leaving the RNA, allowing it to move more freely. However, unlike puromycin, cycloheximide stabilizes ribosomes on mRNA, freezing them in place. It's therefore unclear how this would be expected to make RNA apparently more mobile.

We thank reviewer 1 for this helpful comment. As correctly noted by reviewer 1, cycloheximide is commonly used to block translation elongation by disrupting translocation of A- and P-site tRNAs (by binding to the E site of the large subunit) which stabilises polysomes on RNA transcripts, leaving the RNA in an exposed confirmation.

However, the effects of cycloheximide on translation elongation is most prominent only during a very short time window after drug addition (Buchanan et al., 2016; Hanna et al., 2003)(Smith et al., 2021). Thus, we now provide new data using a shorter incubation time (5 min) which demonstrates that cycloheximide results in a slight reduction in RNA speed compared to controls (new bars in Fig. 4r) which is in line with previous work (Dave et al., 2023; Horvathova et al., 2017).

After a prolonged incubation time (equal or > 2 h), cycloheximide-induced ribosome stabilisation is diminished which may explain the differences observed between for 5-10 min and 2 hour cycloheximide incubation times. We speculate that, for instance, as the cells recover from a period of translation inhibition, they may undergo a period of 'catch-up' translation. Deciphering the mechanism of action of cycloheximide in the context of the living mouse embryo is certainly an interesting area of future research.

Collectively, we see a more pronounced influence on RNA foci speeds in apical regions compared to basal regions, supporting our previous results of increased translation rates and translation component recruitment apically.

4. In figure 5, the authors argue that the observed RNA asymmetry contributes to embryo development as disturbing the asymmetry using ANXA11 knockdown perturbs development. This may be true, but in addition to disturbing RNA distribution, ANXA11 knockdown also disturbed lysosome distribution (extended data fig 3D), so it's difficult to directly ascribe observed effects only to perturbing asymmetric RNA inheritance.

We thank reviewer 1 for this comment. To distinguish if RNA or lysosome asymmetries were causing the observed impaired embryonic development, we performed two additional experiments:

- 1) Overexpression of ANXA11-R346C mutants in living preimplantation embryos to reduce specifically the association of ANXA11 with RNA (Liao et al., 2019; Lillebostad et al., 2020)). In our new Fig. 3j and Supplementary Fig. 3e, f, we show that in this experimental setup RNA remains apically but lysosomes accumulate basally. Similar to the ANXA11 siRNA experiment, embryonic development is delayed at the 16- to 32-cell stage (new Fig. 3i).
- 2) Downregulation of BLOC-1 Related Complex subunit 7 (Borcs7), which is part of the Borcs complex driving microtubule plus end-directed localisation of lysosomes (Pu et al., 2015). Contrary to the experiment above, RNA was able to move basally in Borcs7 siRNA-microinjected embryos (Fig. 3j and Supplementary Fig. 3e). Lysosomes, however, were not asymmetrically localised but instead widely distributed within the cellular volume of late 16-cell stage outer blastomeres (Supplementary Fig. 3f). Despite the effect on lysosome localisation, the embryos developed without any significant delay to 32-cell stage (new Fig. 3i).

Thus, our new data demonstrate that disturbed RNA asymmetries but not lysosome asymmetries are the main driver for developmental delays of preimplantation mouse embryos from 16- to 32-cell stage.

MINOR COMMENTS

1. In line 149, the authors say “Directed transport of RNA cannot occur by diffusion but requires the guidance of the microtubule cytoskeleton...” This is not quite true. For example, Nos RNA in the *Drosophila* embryo is localized through cytoplasmic streaming / diffusion and then anchoring at the site of localization. In yeast Ash1 RNA is localized to daughter cells through an actin-dependent, not microtubule-dependent, system. There even need not be directed transport at all to establish RNA gradients. In *Drosophila* embryos, Hsp83 RNA gradients are established through spatially defined regions of RNA protection and degradation.

We amended the sentence accordingly.

2. There are places where the wording overstates conclusions that can be drawn. For example, in line 165, the authors say that “As the embryos progresses from early to mid 16-cell stage the microtubule network expands, and at the same time the RNA moves basally, indicating a highly spatiotemporally regulated microtubule-dependent transport of RNA at the time of the first embryonic cell fate decision.” These two findings alone do not indicate that. They simply indicate that these two things are happening at the same and therefore may be connected. To directly indicate microtubule-dependent transport, the system should be specifically perturbed, which the authors do later with Ciliobrevin, but not here.

We amended the section accordingly.

Reviewer #2 (Remarks to the Author):

The paper by Zenker and colleagues examines the distributions of apicobasal RNA and its role in cell fate in the early mouse embryo. The study relies heavily on the usage of the RNA-selective fluorescent dyes F22 and E36 that were synthesized for visualizing RNA. The caveat for RNA-selective dyes in general is that it is unclear what type of RNA is being looked at because the dye may label rRNA, tRNA, or even mitochondria RNAs (their mRNAs, rRNA, tRNA, etc.). As the focus of this study presumably is on nuclear-encoded cytosolic mRNA, the authors need to employ another method to verify that the fluorescent puncta they are studying are actually mRNA. For example, they compared the ER density with the presence of Cdx2 and SOX2, so smFISH of mRNAs controlled by these factors could be done.

We thank reviewer 2 for this very helpful comment, which is in line with the first comment of reviewer 1. As outlined there, we performed poly(A) RNA FISH, tFISH and rRNA-naphthalimide live imaging on 16-cell stage preimplantation embryos to determine the mRNA, tRNA and rRNA content of RNA-F22 labelled signals (new Fig. 4a-f, Supplementary Fig. 4a-d). We kindly refer to our reply for reviewer 1, question 1 for more details. Briefly, RNA-F22 signals can contain mRNA and tRNA, in addition to rRNA, which occurs significantly more often in apical regions. In context of translation capacity, we extended these studies with an RNA FISH experiment for eIF2 β mRNA which was detected in apical F22-RNA labelled foci in outer blastomeres of 16-cell stage embryos, in accordance with its protein expression.

We gratefully followed the advice from reviewer 2 and adopted our RNA FISH experiments to two single RNA transcripts, controlled by Cdx2 and Sox2, onto 32-cell stage embryos.

To identify potential candidate targets of Sox2 and Cdx2 during the 16-cell and 32-cell stage we analysed Chromatin immunoprecipitation sequencing (ChIP-seq) data. As currently there are no ChIP-seq data sets for Sox2 or Cdx2 at the 16-cell and 32-cell stages, we utilised data from epiblast stem cells (EPI) (where Sox2 is a known master regulator in pluripotency/inner cells), and trophoblast like cells (TSL) (due to the known roles of Cdx2 in trophoblast development) (Adachi et al., 2013).

Accordingly, in the cells where a high number of binding sites are expected for Sox2 (EPI) and Cdx2 (TSL), we were able to identify most likely potential targets, which could also be regulated by these proteins during preimplantation development. We identified shared and unique target regions across the cell types for each protein (new Supplementary Fig. 5c-f). We then utilised the (Boroviak et al.) scRNA-seq dataset to identify marker genes for several preimplantation stages, which were also targets of either Cdx2 or Sox2 (new Supplementary Fig. 5e). We identified Tfap2c to be a marker around the 8- to 16- cell stage, and Nanog was a marker of the early ICM, which were targets of Cdx2 and Sox2. In addition, upon observing the expression of these two candidate genes, Sox2 and Cdx2, we identified that Cdx2 and Tfap2c follow a unique expression profile throughout the preimplantation developmental period, distinct from the shared Sox2 and Nanog expression pattern (new Supplementary Fig. 5e). Finally, we observed the binding profiles of these peaks at the target regions shown in new Supplementary Fig. 5c, d.

Thus, we selected Tfap2c and Nanog as our candidate mRNA targets regulated by Cdx2 and Sox2. To verify the fluorescent puncta, labelled by the RNA-F22 probe are mRNA we performed RNA FISH for Tfap2c and Nanog in 32-cell stage embryos, as suggested by reviewer 2 (new Supplementary Fig. 5g,h). These results show that the RNA-F22 probe co-localises with the RNA FISH signal.

In conclusion, we show that the RNA-F22 probe co-labels with the RNA FISH signal against specific mRNA transcripts known to be predicted to be regulated by Cdx2 and Sox2 in the early mouse embryo.

The evidence provided for increased translation is inadequate. Their evidence of translation is the accumulation of factors required for translation (i.e. ribosomal protein L10A, PolyA binding protein, eIF2beta subunit, and ER) rather than the actual translation itself. To my eyes, these factors seem to surround the “granule-looking” RNA signals rather than being co-localized within the granule. This would suggest the granules are devoid of translation factors. Even if co-localized in stress granules, some ribosomal subunits and other initiation factors and PABPs may associate with the granule but not necessarily engage in translation. To demonstrate that translation is occurring in granules, assays showing translation using anti-puromycylation after puromycin are recommended.

We agree with the reviewer in the importance of showing direct evidence of translation and greatly appreciate the suggestion of performing the ribopuromycylation assay (RPM assay). We successfully adopted the RPM protocol (Bastide et al., 2018; David

et al., 2012; David et al., 2011) on fixed preimplantation mouse embryos to visualise global translation events. In our new Figure 4n, o and Supplementary Fig. 4f, g, we show a higher number of ribosome-attached puromycin-labelled nascent peptides in apical regions of 16-cell stage outer blastomeres. Moreover, the puromycin signal co-localised with apical RNA F22-labeled foci in outer blastomeres of late 16-cell stage embryos (Figure 4p, q and Supplementary Fig. 4h).

To further verify these findings, we decided to also apply the SunTag system (Tanenbaum et al., 2014; Yan et al., 2016). This system allowed us to monitor translation events in real-time of a specific gene. In context of our discovery, we used the SunTag reporter to visualise the eukaryotic translation initiation factor eIF2 β (new Fig. 4s-u and Supplementary Fig. 4i-p). Coinciding with its mRNA and protein distribution, also the translation of eIF2 β mRNA occurs at apical RNA foci, fortifying the need of rapid post-transcriptional gene expression regulation of translation components in the outer blastomeres of 16-cell stage embryos.

In conclusion, our new experiments show direct translation events at apical RNA foci in fixed (RPM) and live (SunTag) embryos.

The role of annexin A11 for RNA trafficking to the basal location of the outer blastomere, whose motion appears to be dependent on microtubules, is exciting. To demonstrate the RNA signal in these experiments represent RNA granules, the authors need to stain for G3BP1 and/or DBP1.

We thank Reviewer 2 for this suggestion. RNA granules serve as a site of RNA storage, translational or degradation control in many types of cells (Ivanov et al., 2019). However, the presence, identity, and composition of RNA storage sites in mammalian embryos, called embryonic RNA granules, is much less understood. RNA granules, including P-body structures, disappear during mammalian oocyte maturation (Flemr et al., 2010). A recent study (Cheng et al., 2022), showed that maternal mRNAs are stored at mitochondria in mature mouse oocytes. By performing live imaging of eGFP-G3BP1 in mouse embryos at 16-cell stage, we show that typical RNA granules are still absent during preimplantation embryogenesis. We have included these data in the main text stating “We did not observe a similar spatiotemporal distribution pattern or co-localisation with Rab11a-GFP or eGFP-G3BP1, markers for early and recycling endosomes or RNA granules (Supplementary Fig. 3b, c), thus reasoning LAMP1-labelled vesicular structures are most likely lysosomes.” and in Supplementary Fig. 3c. Given that the RNA storage sites during mammalian preimplantation development are still unknown, the identification of them at this time is beyond the scope of the current manuscript.

With additional work to address the points above, the paper can be made suitable for publication in Nature Communications.

Thank you very much. We sincerely hope that we have satisfactorily answered your comments.

References

- Adachi, K., Nikaido, I., Ohta, H., Ohtsuka, S., Ura, H., Kadota, M., Wakayama, T., Ueda, H. R. and Niwa, H.** (2013). Context-dependent wiring of Sox2 regulatory networks for self-renewal of embryonic and trophoblast stem cells. *Mol Cell* **52**, 380-392.
- Bastide, A., Yewdell, J. W. and David, A.** (2018). The RiboPuromylation Method (RPM): an Immunofluorescence Technique to Map Translation Sites at the Sub-cellular Level. *Bio-protocol* **8**, e2669.
- Boroviak, T., Stirparo, G. G., Dietmann, S., Hernando-Herraez, I., Mohammed, H., Reik, W., Smith, A., Sasaki, E., Nichols, J. and Bertone, P.** (2018). Single cell transcriptome analysis of human, marmoset and mouse embryos reveals common and divergent features of preimplantation development. *Development* **145**.
- Cheng, S., Altmeppen, G., So, C., Welp, L. M., Penir, S., Ruhwedel, T., Menelaou, K., Harasimov, K., Stützer, A., Blayney, M., et al.** (2022). Mammalian oocytes store mRNAs in a mitochondria-associated membraneless compartment. *Science* **378**, eabq4835.
- David, A., Dolan, B. P., Hickman, H. D., Knowlton, J. J., Clavarino, G., Pierre, P., Bennink, J. R. and Yewdell, J. W.** (2012). Nuclear translation visualized by ribosome-bound nascent chain puromylation. *Journal of Cell Biology* **197**, 45-57.
- David, A., Netzer, N., Strader, M. B., Das, S. R., Chen, C. Y., Gibbs, J., Pierre, P., Bennink, J. R. and Yewdell, J. W.** (2011). RNA binding targets aminoacyl-tRNA synthetases to translating ribosomes. *J Biol Chem* **286**, 20688-20700.
- Flemr, M., Ma, J., Schultz, R. M. and Svoboda, P.** (2010). P-body loss is concomitant with formation of a messenger RNA storage domain in mouse oocytes. *Biol Reprod* **82**, 1008-1017.
- Ivanov, P., Kedersha, N. and Anderson, P.** (2019). Stress Granules and Processing Bodies in Translational Control. *Cold Spring Harb Perspect Biol* **11**.
- Tanenbaum, M. E., Gilbert, L. A., Qi, L. S., Weissman, J. S. and Vale, R. D.** (2014). A protein-tagging system for signal amplification in gene expression and fluorescence imaging. *Cell* **159**, 635-646.
- Yan, X., Hoek, Tim A., Vale, Ronald D. and Tanenbaum, Marvin E.** (2016). Dynamics of Translation of Single mRNA Molecules In Vivo. *Cell* **165**, 976-989.

REVIEWERS' COMMENTS

Reviewer #1 (Remarks to the Author):

With a considerable amount of work, the authors have addressed my concerns, although I still have some lingering reservations regarding interpretations of F22 experiments. Nevertheless, I congratulate them on strengthening their conclusions and greatly improving the manuscript.

I have one remaining comment that I want to point out. In point 2 of my original review, I had noted that using a dynein inhibitor to monitor plus-end directed transport is quite indirect and that it would be better to instead inhibit kinesin-based transport. In their rebuttal, the authors state why this is not possible. Some of their points raised here are incorrect, particularly those regarding kinesore.

The authors state that kinesore is not a viable option here because it activates, not inhibits, kinesin-1. It is true that kinesore activates kinesin-1 activity. However, it does so in a way that prevents transport of cargo by kinesin-1. This is shown in the very paper that the authors use as evidence for why kinesore cannot be used (Randall et al, 2017). Further, kinesore has been used to inhibit kinesin-based RNA transport to the basal end of epithelial cells (Goering et al, bioRxiv, 2022), a very similar experiment to what the authors are trying to show here. Observing RNA localization defects in response to kinesore treatment would **greatly** strengthen the authors' claims regarding plus-end directed transport.

Reviewer #2 (Remarks to the Author):

The authors have addressed my concerns appropriately.

Response to reviewers' comments

Reviewer #1 (Remarks to the Author):

With a considerable amount of work, the authors have addressed my concerns, although I still have some lingering reservations regarding interpretations of F22 experiments. Nevertheless, I congratulate them on strengthening their conclusions and greatly improving the manuscript.

I have one remaining comment that I want to point out. In point 2 of my original review, I had noted that using a dynein inhibitor to monitor plus-end directed transport is quite indirect and that it would be better to instead inhibit kinesin-based transport. In their rebuttal, the authors state why this is not possible. Some of their points raised here are incorrect, particularly those regarding kinesore.

The authors state that kinesore is not a viable option here because it activates, not inhibits, kinesin-1. It is true that kinesore activates kinesin-1 activity. However, it does so in a way that prevents transport of cargo by kinesin-1. This is shown in the very paper that the authors use as evidence for why kinesore cannot be used (Randall et al, 2017). Further, kinesore has been used to inhibit kinesin-based RNA transport to the basal end of epithelial cells (Goering et al, bioRxiv, 2022), a very similar experiment to what the authors are trying to show here. Observing RNA localization defects in response to kinesore treatment would **greatly** strengthen the authors' claims regarding plus-end directed transport.

We thank Reviewer #1 for taking the time to review our manuscript again and for their comment. As described by Reviewer #1, kinesore activates Kinesin-1 movement towards microtubule plus ends but, at the same time, blocks transport due to its binding to the cargo-binding interface of Kinesin-1 (Randall et al., 2017).

The suitability of kinesore for our work, however, is limited due to the following reasons:

- Kinesore causes bundling and sliding of microtubule filaments and thus, the entire microtubule network becomes significantly disorganised (Randall et al., 2017). If treating living 16-cell stage embryos with high kinesore concentrations, required for kinesore functionality (Randall et al., 2017), it might remain unclear if potential defects in RNA localisation may be caused due to Kinesin-1-dependent phenotypes or the overall disorganisation of the microtubule network.
- It is excellent to see that kinesore was successfully applied to specifically inhibit the kinesin-1-dependent transport of RNA to basal regions (Goering et al., 2023). In differentiated enterocyte-like C2bbe1 cells, however, RNA transport is facilitated by RNA-binding proteins (RBPs) (Goering et al., 2023) which is unlikely to occur in preimplantation mouse embryos due to the absence of RBP structures (Supplementary Fig. 3c of our manuscript). Instead, RNA hitchhikes on lysosomes, anchored by Annexin A11, in the cells of the early embryo (Fig. 3 of our manuscript). A multitude of kinesins might be involved in the transport of lysosomes, including Kinesin-1, Kinesin-2, Kinesin-3 and Kinesin-13 (Pu et al., 2015). Thus, kinesore treatment might only cause a partial defect requiring an extensive characterisation if this is caused by 1) the activity of other kinesins, 2) insufficient penetration of high concentrations of kinesore through the larger 3-dimensional embryo or 3) basal RNA transport being independent of Kinesin-1 in the preimplantation embryo.

We appreciate Reviewer #1's comment and have amended our manuscript as followed to reflect the indirect proof using Ciliobrevin D, blocking microtubule minus end directed transport:

Page 7 (Results): "These results suggest that microtubule plus end transport might be triggering the accumulation of basal RNA clusters. Furthermore, microtubule minus end transport might be directed apically and responsible for the retention of the smaller apical RNA foci."

Page 15 (Discussion): removed “via microtubule plus end-directed transport”

Figure 2j, left: removed “plus end directed transport” in both title and legend by amending to “Microtubule-dependent RNA transport” (title) and “Schematic representation of polarised microtubules directing RNA transport” (legend).

Reviewer #2 (Remarks to the Author):

The authors have addressed my concerns appropriately.
We thank Review #2 for the time taken to review our manuscript.

REFERENCES

Goering, R., Arora, A., Pockalny, M. C. and Taliaferro, J. M. (2023). RNA localization mechanisms transcend cell morphology. *Elife* 12, e80040.

Pu, J., Schindler, C., Jia, R., Jarnik, M., Backlund, P. and Bonifacino, J. S. (2015). BORC, a multisubunit complex that regulates lysosome positioning. *Developmental cell* 33, 176-188.

Randall, T. S., Yip, Y. Y., Wallock-Richards, D. J., Pfisterer, K., Sanger, A., Ficek, W., Steiner, R. A., Beavil, A. J., Parsons, M. and Dodding, M. P. (2017). A small-molecule activator of kinesin-1 drives remodeling of the microtubule network. *Proc National Acad Sci* 114, 13738-13743.